# Single-Pass Document Scanning for Question Answering

**Weili Cao,**[*] **Jianyou Wang,**[*] **Youze Zheng,**[*] **Longtian Bao,**[*] **Qirui Zheng**
**Taylor Berg-Kirkpatrick, Ramamohan Paturi,**[†] **Leon Bergen**[†]
Laboratory for Emerging Intelligence
University of California, San Diego
{w2cao,jiw101,yoz018,lobao}@ucsd.edu

## Abstract

Handling extremely large documents for question answering is challenging: chunk-based embedding methods often lose track of important global context, while full-context transformers can be prohibitively expensive for hundreds of thousands of tokens. We propose a single-pass document scanning approach that processes the entire text in linear time, preserving global coherence while deciding which sentences are most relevant to the query. On 41 QA benchmarks, our single-pass scanner consistently outperforms chunk-based embedding methods and competes with large language models at a fraction of the computational cost. By conditioning on the entire preceding context without chunk breaks, the method preserves global coherence, which is especially important for long documents. Overall, single-pass document scanning offers a simple solution for question answering over massive text. All code, datasets, and model checkpoints are available at https://github.com/MambaRetriever/MambaRetriever

## 1 Introduction

Long document question answering remains a significant challenge in natural language processing due to the quadratic computational cost associated with using transformer-based Large Language Models (LLMs) (Vaswani et al., 2017). Retrieval-Augmented Generation (RAG) (Asai et al., 2024) addresses this issue by processing long documents in shorter chunks with embedding models, thereby maintaining an approximately linear computational cost relative to context size. These embedding models then retrieve relevant chunks to serve as input for an LLM to generate an answer. However, embedding models that only process shorter chunks may lose important global and contextual information. This limitation has spurred ongoing research into context-aware embedding models (Morris & Rush, 2024).

In this paper, we introduce a new approach: Single-Pass Document Scanning. The Single-Pass Scanner is a state-space model (SSM) (Dao & Gu, 2024) that processes long documents in its entirety with linear scaling in sequence length. It uses its understanding of the entire preceding context to identify relevant sentences, as illustrated in Figure 1.

The Single-Pass Scanner outperforms state-of-the-art embedding models across 41 long-document QA benchmarks (see Table 1). It also maintains faster speed at processing documents and uses fewer FLOPs than embedding models while achieving higher accuracy (see Table 2). The Single-Pass Scanner generalizes well on very long documents, achieving performance close to GPT-4o's (OpenAI, 2024) full-context capabilities on documents longer than 256k tokens (see Figure 3), while only using 1600 tokens from 50 relevant sentences it identifies.

Another key contribution of this paper is our link-based synthetic data generation method, which identifies connections within a document and transforms these connections into questions that require using different parts of the document to answer. When trained

---

[*]equal contributions
[†]equal senior authors

with our synthetic data, Single-Pass Scanners can leverage long-range connections within documents to more accurately identify relevant sentences[1]. Our experiments demonstrate the superiority of our link-based method over baseline synthetic generation methods (see Table 4).

## 2 Related Work

**Long-context Language Models:** Transformer models are inefficient when processing long-context documents because they suffer from quadratic scaling in both training and inference (Liu et al., 2024a). Many works are dedicated to reducing transformer's quadratic complexity while improving global reasoning in long documents. Sparse-attention models such as Longformer (Beltagy et al., 2020) and LongT5 (Guo et al., 2022) achieve linear scaling at the expense of some performance degradation. Other work focuses on using customized synthetic data and architectures to effectively extend the context window size of language models (Zhang et al., 2024c; An et al., 2024b; Luo et al., 2024; Xiong et al., 2024).

**State Space Models:** Meanwhile, SSMs emerge as an alternative to process long sequences (Fu et al., 2024; Gu et al., 2022; Peng et al., 2023; Arora et al., 2024), as they have linear scaling during training and inference. Dao & Gu (2024) incorporated input-dependent parameters into SSMs and integrate efficient parallelizable training and efficient autoregressive inference. Glorioso et al. (2024) and Waleffe et al. (2024) proposed a hybrid Mamba that combines Mamba with attention. Arora et al. (2024) used non-causal prefix-linear-attention to improve model understanding of the global context. Some recent works built embedding models from SSMs (Hwang et al., 2024; Zhang et al., 2024a), for example, the M2-BERT (Saad-Falcon et al., 2024) embedding model from the Monarch Mixer architecture (Fu et al., 2024).

**Retrieval-Augmented Generation (RAG):** The approach of retrieving information using an embedding model followed by generating an answer has been fundamental for processing long texts that exceed the context limits of language models (Nakano et al., 2021; Borgeaud et al., 2022; Wang et al., 2023a; Izacard & Grave, 2020; Huang et al., 2023; Liu et al., 2024b; Xu et al., 2024b; Yu et al., 2024).

**Transformer-based Embedding Models:** Transformer-based embedding models are typically used as retrievers for RAG systems. Previous embedding models focus on semantic understanding and instruction-following (OpenAI, 2024; Wang et al., 2023b; Izacard et al., 2021; Lin et al., 2023; BehnamGhader et al., 2024). Embedding models NV-Embed-v2-7B (Lee et al., 2024), GTE-Qwen2 (Li et al., 2023), Stella (Zhang, 2024) and GritLM (Muennighoff et al., 2024) excel at identifying semantically similar sentences within localized contexts (Muennighoff et al., 2023). Since embedding models suffer from the lack of contextual understanding, Morris & Rush (2024) proposed context-aware embedding models. Xu et al. (2024a); Huang et al. (2024) utilize transformer-based embeddings in order to shrink the retrieval context provided to downstream LLMs.

**Our work differs from transformer or SSM based embedding models.** While most embedding models select text chunks without awareness of global context, and even context-aware embedding models can only see limited neighborhood context, our Single-Pass Scanner identifies relevant sentences based on the entire preceding context in a single pass. This enables our model to maintain context when processing extremely long documents.

## 3 Methodology

### 3.1 Preliminaries

We describe the model architecture of SSM. We denote the head dimension as $P$, and the state expansion factor as $N$. We give the recursion formula of a 1-dimensional sequence mapping $x_t \in \mathbb{R} \mapsto y_t \in \mathbb{R}$ through an implicit latent state $h_t \in \mathbb{R}^N$, where parameters $A \in \mathbb{R}^{N \times N}, B \in \mathbb{R}^{N \times 1}, C \in \mathbb{R}^{N \times 1}$.

---

[1]See examples in Appendix E.7 where the Single-Pass Scanner uses contextual information.

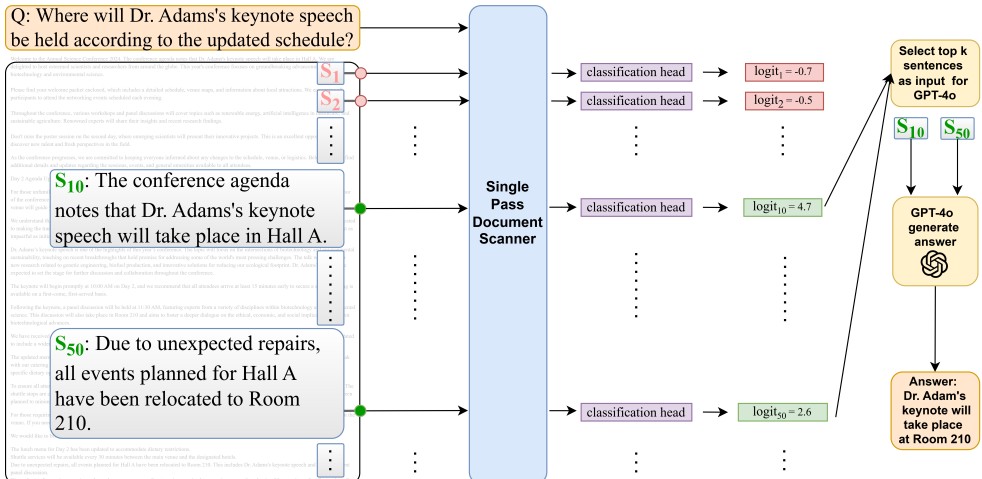

Figure 1: Documents may have long-range dependencies useful for answering questions. On the left, $S_{50}$ is relevant to $Q$ only through its dependency on $S_{10}$. On the right, our Single-Pass Scanner uses a classification head on the last token of each sentence to generate logits. Sentences with the top-k logits are selected for an LLM to generate an answer.

$$h_0 = Bx_0 \quad \ldots \quad \begin{cases} h_t = Ah_{t-1} + Bx_t \\ y_t = C^T h_t \end{cases} \quad \ldots \quad \begin{cases} h_T = Ah_{T-1} + Bx_T \\ y_T = C^T h_T \end{cases}$$

The equation above defines a sequence transformation for $P = 1$, and it can be generalized to $P > 1$ for $x_t, y_t \in \mathbb{R}^P$ by broadcasting across this dimension.

### 3.2 Model Architecture of Single-Pass Scanner

Single-Pass Scanner is built with the Mamba-2 architecture (Dao & Gu, 2024). From the pretrained Mamba-2 checkpoint, we remove the language modeling head and replace it with a classification head. We denote the binary classification head as $H \in \mathbb{R}^{P \times 1}$, and logit $z_t$ can be computed as $z_t = y_t H$. The end of sentence logits $z_{s_1}, z_{s_2}, \ldots, z_{s_n}$ represent model's judgment of each sentence's relevant to the query.

To apply the Single-Pass Scanner to a specific query and document, we first concatenate the query $Q$ and document $D$. This combined text is then tokenized into a sequence of tokens $u_0, u_1, \ldots, u_T$, where $T$ represents the time axis. We denote the index of the last token of each sentence as $s_1, \ldots, s_n$ where $0 < s_i \leq T$. Note $s_n = T$. The input list of tokens $u_0, \ldots, u_T$ are projected to latent space as $x_0, \ldots, x_T$ where $x_t \in \mathbb{R}$.

**During training**, we are given $n$ binary relevance labels $r_1, r_2, \ldots, r_n$ corresponding to the $n$ input sentences, where $r_i \in \{0, 1\}$. These labels indicate whether each sentence is relevant to the query. The end of sentence logits $z_{s_1}, z_{s_2}, \ldots, z_{s_n}$ with corresponding labels $r_1, r_2, \ldots, r_n$ are used to compute the cross-entropy loss, $\sum_{i=1}^{n} -w_i \left[ (r_i \log z_{s_i} + (1 - r_i) \log(1 - z_{s_i}) \right]$, where $w_i$ is a data-dependent weight to upsample the number of positive labels. For more details on the construction of training data, refer to Section 4.1.

**During inference**, the model processes $Q + D$ in a single pass. Based on the logit value of the last token in each sentence, we select the top-k sentences to input into a generator model, such as GPT-4o, for final answer generation (see Figure 1). Selection decisions for each sentence are conditioned on all prior tokens in the document, preserving global context.

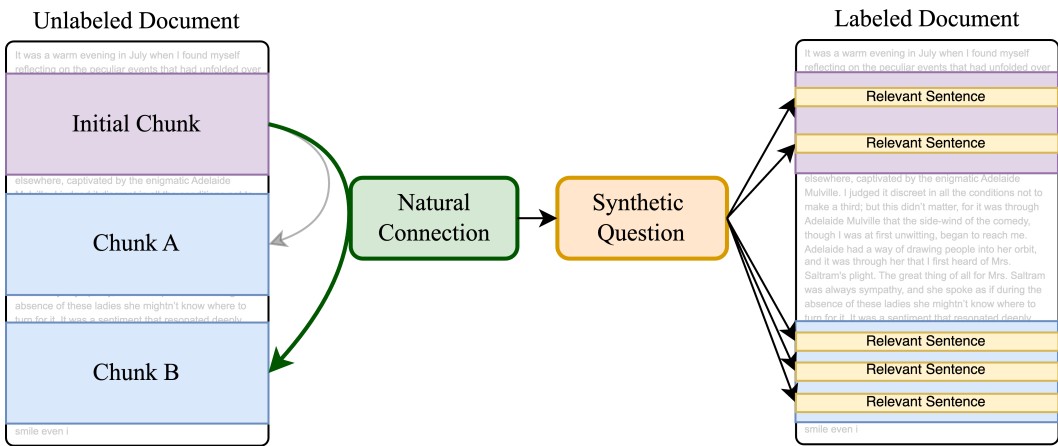

Figure 2: The link-based synthetic data generation strategy. This approach first uses an LLM to identify natural connections between different chunks of the document; in the diagram, the initial chunk has a connection to Chunk B but not Chunk A. The LLM then generates a question related to this connection. Finally, the LLM assigns a binary relevance label to each sentence in the two chunks.

# 4 Synthetic Data Generation

As mentioned in Section 3, training the Single-Pass Scanner requires labeled long-document data, where each sentence is tagged as relevant or irrelevant to a query. Although large corpora of long documents exist, they typically lack sentence-level relevance labels. To address this gap, we propose a synthetic data generation framework that (1) formulates questions from real documents and (2) assigns binary relevance labels to individual sentences. This section describes our new link-based strategy and contrasts it with two baselines.

**Chunk-based generation** involves selecting a local passage from the document and crafting a question around that passage alone. First, a random segment of 20 consecutive sentences is sampled from the document. An LLM is then instructed to generate one synthetic question specific to this segment and to label each sentence in the segment for its relevance. Although this method captures local context, it cannot produce questions requiring long-distance reasoning. The chunk-based approach overlooks factual connections that span multiple parts of the document. See Figure 5 in Appendix B.1 for prompts.

**Pair-based generation** involves generating question based on two different parts of the document. Here, the text is divided into distinct 20-sentence chunks, each of which is embedded with a sentence-level embedding model. The chunk most similar to a randomly chosen "query chunk" is identified through cosine similarity in the embedding space, and an LLM is prompted to produce a question that requires information from both chunks. Although this method tries to elicit reasoning over distant pieces of text, it sometimes pairs chunks that share superficial lexical similarities rather than a coherent logical connection. The resulting questions may lack contextual consistency. For detailed prompts used in this pair-based generation approach, please refer to Figure 6 in Appendix B.2.

## 4.1 Link-based Generation

The link-based generation strategy improves on pair-based strategy and produces more realistic and coherent questions by leveraging natural connections between different sections of a document. An LLM is initially presented with one random 20-sentence chunk and is then provided with a 200-sentence context window that includes the initial chunk at the beginning or the end. The model identifies meaningful dependencies (e.g., the thematic reappearance of a character) of the initial chunk with another chunk inside the window. This process is not recursive. Once it identifies a relevant link and this other chunk, it is asked to

generate a question that depends on both chunks. The LLM then outputs a list of chunks along with their connections to the initial chunk. As illustrated in Figure 2, a synthetic question is generated based on a natural connection, and relevant sentences are labeled. Because it relies on textual relationships rather than purely semantic embedding scores, the link-based method tends to generate more coherent training examples. See Appendix B.3 for prompts used.

GPT-4o-mini-2024-07-18 is the LLM used for synthetic data generation. See Table 4 for the financial cost of each strategy. See Appendix B.4 for examples of synthetic data generated from each strategy and human evaluation of these synthetic examples.

### 4.2 Data Sources for Synthetic Data Generation

The synthetic generation pipeline used long documents which were collected from Project Gutenberg (2024), government reports dataset (Huang et al., 2021), finance documents from U.S. Securities and Exchange Commission (2024), and legal contracts (Hendrycks et al., 2021). Synthetic data examples were generated by selecting random subsequences ranging from 2k to 10k tokens from a long document.

### 4.3 Decontamination from Test Sets

To prevent contamination between our training and testing data, we implemented the following procedure. First, we divided all documents from our 41 test sets into individual sentences, yielding a set of 2.4 million test sentences. Next, we split the document from each synthetic data point into sentences and calculate the overlap with the test set using string matching. We removed any synthetic data point where more than 1% of its sentences matched those in the set of 2.4 million test sentences. This process effectively eliminated textual overlap between the test and training sets.

## 5 Experimental Methods

### 5.1 Test Sets & Validation Sets

For testing, evaluation is done on 41 QA benchmark test sets from Bai et al. (2024); Zhang et al. (2024b); An et al. (2024a); Yuan et al. (2024); Dong et al. (2024); Thonet et al. (2024); Reddy et al. (2024); Hudson & Al Moubayed (2022). For clarity of presentation, based on document types reported in their original sources, we categorize all 5735 data points from these 41 test sets into 4 categories: educational, creative, official and conversational. Details of these 4 categories are provided in Appendix A.1. For evaluation metrics, we report per-category accuracy and average accuracy across all data points.

**Validation Sets** are taken from the train sets of 8 benchmark tasks, and are only used for hyperparameter tuning. We verified validation sets are completely disjoint from test sets. See details of test sets and validation sets in Appendix A.1, A.2, A.3.

### 5.2 Evaluated Systems

**Full-context LLMs:** LLMs such as GPT-4o and Llama 3.1 (AI@Meta, 2024) process the entire document in-context and answer the question directly. We also fine-tuned Mamba-2 for full context answer generation in Appendix E.3.

**Single-Pass Scanner:** Single-Pass Scanners process the full document in-context, and select the top 50 relevant sentences for an LLM generator. In the Appendix A.4, we also report another setting where Single-Pass Scanners select the top 10 sentences.

**RAG with embedding models:** For fairness, we consider two setups. The **"5 chunks"** setup follows the standard RAG setup (Xu et al., 2023; Li et al., 2024) where the documents are processed in chunks of fixed-length 300 words, and embedding models retrieve the top 5 chunks. The **"50 sentences"** setup matches Single-Pass Scanner's approach. The "5

chunks" setup always achieves higher performance for embedding models (e.g., Dragon (Lin et al., 2023)), and BM25 (Trotman et al., 2014; Robertson & Walker, 1994) (See Table 2). Therefore, we only report the "5 chunks" results for both embedding models and BM25 in the main paper for brevity. In Appendix A.4, we also report the "50 sentences" setup for all embedding models for completeness.

## 5.3 GPT-4o as Judge

The accuracy of freeform answers is evaluated using GPT-4o-2024-08-06, which uses a specialized prompt to compare attempted answers with ground-truth answers, providing a binary "yes" or "no" judgment. The prompt is developed from 100 human-annotated examples. On a separate held-out test set of 180 human-annotated examples, GPT-4o's 180 yes/no judgments have a high agreement with human judgments, achieving a 0.942 macro F1 score. See Appendix C for the prompt.

## 5.4 Sliding Window

LLMs like GPT-4o have a context length limit of 128k tokens. To standardize evaluation on full-context LLMs, we employ a sliding window approach for documents exceeding 120k tokens. This approach uses a window size of 120k tokens and a stride of 60k tokens. The answers from different windows are then aggregated by the same LLM, which then produces a final answer.

Our Single-Pass Scanner can generalize beyond its training context length of 10k tokens (see Section 7.2). For instance, the SPScanner 1.3B can handle up to 256k tokens without memory errors on a single node with $8 \times$ 80GB H100. To ensure a fair comparison with GPT-4o, we use the same sliding window approach for Single-Pass Scanners when documents exceed 120k tokens. This allows both models to operate within the same effective context window. Sentences scored twice have their scores (i.e., logit values) averaged.

## 5.5 Fine-tuning

**Single-Pass Scanners:** From checkpoints in Dao & Gu (2024), the Mamba-2-130M model is fine-tuned on 1 million link-based synthetic data, while the Mamba-2-1.3B model is fine-tuned on 400k data, both for one epoch without early stopping. Due to budget constraints and the lack of additional long-context training documents, we created only 1 million link-based data points. We limited the training of Mamba-2-1.3B to 400k data points because we did not observe any improvements in the validation sets when training beyond this amount.

Learning rates were the only hyperparameters optimized on validation sets. On one node with $8 \times$ 80GB H100s, training the 1.3B model took 5 hours, while the 130M model took 3 hours with their respective training data sizes. See Appendix D for hyperparameter settings.

**Embedding Models:** We fine-tuned two embedding models, Contriever-110M (Izacard et al., 2021) and GTE-Qwen2-1.5B (Li et al., 2023), using the same 1 million link-based synthetic data for one epoch. For each query, relevant sentences are treated as positives and irrelevant sentences are treated as negatives. We used the same contrastive loss and applied the same hyperparameter settings (e.g., scheduler, optimizer, temperature $\tau$ in NT-Xent loss) as reported in their original papers, and we optimized learning rates, batch size, and training data size on the same validation sets.

## 5.6 Document Processing Speed and Efficiency

In Table 2, we evaluate the performance of various retrieval systems using our test sets. Specifically, we measure the average time it takes for a model to process a single long document (already on GPU devices), excluding any pre-processing, post-processing, or host-to-device transfer time. For retrieval systems utilizing embedding models, a long document is processed either in batches of sentences or in chunks of 300 words, depending on the retrieval setting.

For the embedding models, batches consist of sentences or chunks from the same document. The batch size for these models is selected to maximize token throughput. Sentences and chunks of similar lengths are batched together in order to minimize the number of padding tokens. For the Single-Pass Scanners, a batch size of 1 is consistently used, as the entire long document must be processed at once.

When embedding models process input in batches larger than size 1, padding is necessary. Since embedding models require larger batch sizes for faster processing, the additional padding results in higher FLOPs. To provide a more informative comparison, we calculate FLOPs for embedding models both with and without padding. Note that the Single-Pass Scanner does not use padding, so the FLOPs remain the same regardless of padding. FLOPs are calculated using standard formulas provided by Kaplan et al. (2020); Dao & Gu (2024).

Our hardware setup includes two Intel Xeon Platinum 8480+ processors (224 logical CPUs) and 8 × 80GB H100 GPUs.

For embedding models, the "50 sentences" setup retrieves an average of 1600 tokens, while the "5 chunks" setup retrieves an average of 2000 tokens. The "50 sentences" setup consistently results in lower accuracy due to retrieving fewer tokens.

| Retrievers with GPT-4o as Generator | Educational $n = 1967$ | Creative $n = 1733$ | Official $n = 1328$ | Conversational $n = 707$ | Average Accuracy |
|---|---|---|---|---|---|
| BM25 | 62.5 | 37.5 | 46.2 | 41.4 | 49.1 |
| M2-BERT | 52.3 | 27.6 | 37.8 | 34.9 | 38.2 |
| Dragon-110M | 64.9 | 45.1 | 54.1 | 44.6 | 53.9 |
| Contriever-110M | 66.3 | 45.8 | 52.9 | 45.0 | 54.3 |
| Contriever-110M-FT[*] | 65.5 | 48.0 | 55.5 | 41.2 | 54.8 |
| GTE-Qwen2-1.5B | 67.2 | 47.7 | 56.2 | 44.3 | 55.7 |
| GTE-Qwen2-1.5B-FT[*] | 66.9 | 48.0 | 56.2 | 44.8 | 55.8 |
| Stella-1.5B | 66.9 | 50.7 | 54.7 | 47.9 | 56.8 |
| OpenAI v3-large | 68.3 | 50.3 | 57.8 | 48.7 | 57.6 |
| GritLM-7B | 68.3 | 49.7 | 56.2 | 48.7 | 57.2 |
| NV-Embed-v2-7B | 69.7 | 52.7 | 56.3 | 53.2 | 59.1 |
| **SPScanner 130M** | 70.4 | 54.1 | 59.5 | 49.5 | 60.0 |
| **SPScanner 1.3B** | **73.0** | 56.5 | 60.5 | 50.5 | 61.8 |
| GPT-4o Full Context | 71.6 | **62.0** | **62.5** | **62.2** | **64.6** |

Table 1: Single-Pass Scanners select 50 sentences, while BM25 and embedding models retrieve 5 chunks because chunk-based retrieval performed better than sentence retrieval for them. Average Accuracy is calculated across all data points and is not influenced by categories. SPScanner stands for Single-Pass Scanner. FT[*] means fine-tuned. See Section 5.5 for details of fine-tuning models.

## 6    Main Results

**Single-Pass Scanners outperform embedding models:** Table 1 reports model performance grouped by document types and the average performance across all data points, with GPT-4o-2024-08-06 as generator. Both Single-Pass Scanners outperform BM25, all embedding baselines and fine-tuned embedding models, including MTEB (Muennighoff et al., 2023) leaders (as of January 1, 2025), NV-Embed-v2-7B (Lee et al., 2024) and Stella-1.5B (Zhang, 2024). Results on individual dataset performance are provided in Appendix A.4.

**Single-Pass Scanners are computationally efficient and fast at processing documents:** Table 2 compares Single-Pass Scanners with the SoTA embedding models NV-Embed-v2-7B, Stella-1.5B and GTE-Qwen2-1.5B. Section 5.6 explains how speed (i.e., document processing speed) and FLOPs with and without padding are calculated. We see SPScanner 1.3B is slightly faster at processing documents and slightly more computationally efficient than

| Model | Setting | Speed (ms) | TFLOPs | TFLOPs w/o Pad | Params (billions) | Average Accuracy |
|---|---|---|---|---|---|---|
| **SPScanner 130M** | 50 sents | **93.4** | **19.0** | **19.0** | **0.1** | 60.0 |
| **SPScanner 1.3B** | 50 sents | 181.6 | 197.9 | 197.9 | 1.3 | **61.8** |
| NV-Embed-v2-7B | 50 sents | 592.0 | 1316.7 | 1279.4 | 7.9 | 56.6 |
| NV-Embed-v2-7B | 5 chunks | 470.8 | 1295.6 | 1287.5 | 7.9 | 59.1 |
| Stella-1.5B | 50 sents | 364.7 | 331.9 | 210.5 | 1.5 | 55.6 |
| Stella-1.5B | 5 chunks | 264.8 | 244.8 | 219.0 | 1.5 | 56.8 |
| GTE-Qwen2-1.5B | 50 sents | 364.4 | 331.9 | 210.5 | 1.5 | 54.6 |
| GTE-Qwen2-1.5B | 5 chunks | 264.9 | 244.8 | 219.0 | 1.5 | 55.7 |
| Llama-3.1-70B | Direct Answer | N/A | 28,517.9 | 28,517.9 | 69.5 | 57.8 |

Table 2: Section 5.6 explains speed measurements and FLOPs calculations. FLOPs is calculated with padding when the batch size is larger than 1. FLOPs w/o Pad is calculated without padding. Llama-3.1-70B is evaluated as a direct answer generator based on the full context of a long document (see Section E.3). The large FLOPs for Llama-3.1-70B is due to quadratic attention on long sequences.

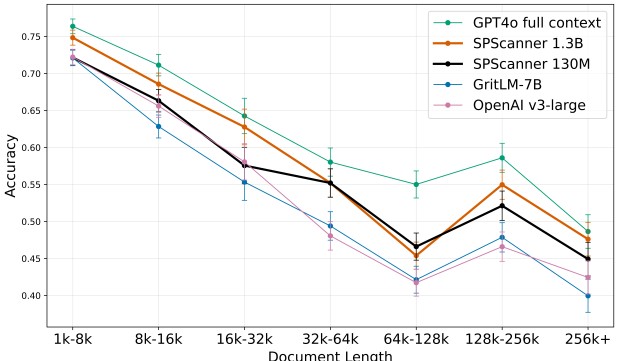

Figure 3: Retrieval models' performance across documents of different lengths with GPT-4o as the generator.

| Model | Llama-3.1 | |
|---|---|---|
| | 70B | 8B |
| BM25 | 46.9 | 39.1 |
| Dragon-110M | 51.9 | 44.1 |
| Contriever-110M | 52.9 | 44.8 |
| OpenAI v3-large | 55.1 | 46.0 |
| GritLM-7B | 55.4 | 44.9 |
| **SPScanner 130M** | 57.5 | 47.4 |
| **SPScanner 1.3B** | **58.8** | **47.9** |

Table 3: Performance of Models paired with different LLMs as generators.

embedding models. SPScanner 130M is considerably faster and uses much fewer FLOPs due to its small size.

**Single-Pass Scanners are robust to different generators:** Table 3 shows average performance when Llama-3.1 8B and 70B are used as the generators. Single-Pass Scanners continue to outperform the embedding-based retrievers in this setting. Appendix A.5 presents results across individual datasets.

**Single-Pass Scanners are comparable to GPT-4o on context over 256k tokens:** Figure 3 shows the performance of the Single-Pass Scanner and other baselines across different document lengths. Single-Pass Scanner shows an increasing advantage over embedding baselines as document length increases, and performance converges to GPT-4o for documents longer than 256k. This shows significant length generalization from the model, which was only fine-tuned on documents up to length 10k.

**Link-based synthetic data is more suitable to train state-space models:** Table 1 shows no improvement on GTE-Qwen-1.5B-FT (i.e., fine-tuned) and Contriever-110M-FT over their pre-trained checkpoints, suggesting Single-Pass Scanners learned more than superficial artifacts such as domain adaptability from training documents. The long-range dependencies in link-based data makes it hard for embedding models with short context windows to learn.

| Strategy | Educational
n = 1967 | Creative
n = 1733 | Official
n = 1328 | Conversational
n = 707 | Average
Accuracy | Cost
($) |
|---|---|---|---|---|---|---|
| Chunk-based | 68.2 | 49.6 | 57.8 | 46.1 | 57.2 | 71 |
| Pair-based | 66.6 | 42.8 | 49.0 | 41.3 | 51.4 | 167 |
| **Link-based** | **69.8** | **51.6** | **59.6** | **50.4** | **59.4** | 1076 |

Table 4: Synthetic data strategies for training Single-Pass Scanner 130M. GPT-4o is generator.

## 7 Ablations

### 7.1 Comparing Synthetic Data Strategies

This paper introduced our novel link-based synthetic generation strategy. We now evaluate its effectiveness against the two other baseline strategies, chunk-based and pair-based generations. We train 130M Single-Pass Scanners under identical experimental conditions with 500k synthetic questions created from the same documents by each of the three strategies. Table 4 shows that the link-based strategy achieves the strongest results. Interestingly, pair-based generation strongly underperforms link-based generation, suggesting flaws in its synthetic questions. Refer to Appendix B.4 Table 21 for some flawed synthetic questions generated from the pair-based strategy. By discovering connections between chunks of text in a document, the link-based strategy is able to generate more coherent and contextually relevant questions that would require information from distinct parts of the document.

### 7.2 Context Size Ablations

To assess whether long-distance context is improving the performance of the Single-Pass Scanner, we perform ablations on the amount of document context provided to the model. In the small context condition, the document is chunked by sentence. In the medium context condition, the document is chunked at 1024 tokens. After chunking, the model processes each chunk independently, and the results are aggregated across chunks.

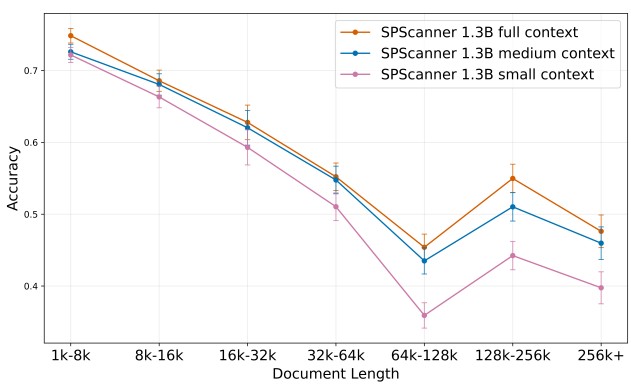

Figure 4: SPScanner Context Size Ablations

Figure 4 shows model performance for the small and medium context ablations with GPT-4o as the generator. As document length increases, the ablated Single-Pass Scanners perform worse relative to the full context setting. This provides evidence that the model is able to make effective use of long-distance context when scanning through the document.

### 7.3 Further Analyses

In Appendix E.2, we investigate the feasibility of using an LLM to generatively select sentences via next token prediction. Results show that generative baselines underperform the discriminative SPScanner. In Appendix E.3, we evaluate the capability of training Mamba-2 to directly answer questions from full document contexts. We found its performance remains substantially lower than that of Single-Pass Scanner. Additionally, in Appendix E.4, we found the increasing distance between linked-chunks and the query improves Single-Pass Scanners. In Appendix E.5, we found longer training document length of up to 10k tokens improves Single-Pass Scanners. In Appendix E.6, we observed Single-Pass Scanners performed slightly worse when important information is located at the end of a long document. In Appendix E.1, we analyze model performance on scientific documents. Model performance on scientific documents are generally consistent with the overall performance.

| Metric | @k | Model | | | | | | |
|---|---|---|---|---|---|---|---|---|
| | | BM25 | Dragon 110M | GTE-Qwen2 1.5B | OpenAI v3-Large | NV-Embed v2-7B | **SPScanner 130M** | **SPScanner 1.3B** |
| Recall | 1 | 11.5 | 12.5 | 14.2 | 15.1 | 15.8 | 16.1 | **19.4** |
| | 5 | 27.3 | 28.7 | 31.5 | 35.2 | 36.8 | 39.6 | **45.5** |
| | 10 | 35.2 | 36.8 | 40.6 | 45.3 | 46.3 | 51.3 | **57.1** |
| | 50 | 56.3 | 59.6 | 64.5 | 68.2 | 69.0 | 74.4 | **79.0** |
| Precision | 1 | 28.3 | 30.7 | 34.0 | 37.4 | 38.7 | 41.0 | **47.8** |
| | 5 | 15.8 | 16.7 | 18.3 | 20.7 | 21.5 | 24.1 | **27.4** |
| | 10 | 11.2 | 11.7 | 12.9 | 14.4 | 14.6 | 17.0 | **19.0** |
| | 50 | 4.4 | 4.7 | 5.0 | 5.3 | 5.4 | 6.2 | **6.6** |
| nDCG | 1 | 28.3 | 30.7 | 34.0 | 37.4 | 38.7 | 41.0 | **47.8** |
| | 5 | 27.6 | 29.3 | 32.2 | 35.8 | 37.3 | 40.5 | **46.7** |
| | 10 | 30.3 | 31.9 | 35.3 | 39.1 | 40.4 | 44.4 | **50.5** |
| | 50 | 36.7 | 38.8 | 42.5 | 46.1 | 47.3 | 51.6 | **57.3** |

Table 5: The information retrieval performances of SPScanners and embedding models on a set of 3,539 data points. The information retrieval benchmark is constructed in Section 8.

# 8 An Information Retrieval Perspective

We can alternatively formulate sentence selection as an information retrieval (IR) problem: given a long document, the goal is to retrieve a small subset of sentences sufficient to answer a query. To construct ground-truth labels for this task, we developed an LLM-assisted annotation–validation pipeline.

In the annotation stage, GPT-4o-mini labeled sentences for relevance. Conditioned on the query and its correct answer, the model processed the document in non-overlapping 20-sentence windows, marking sentences deemed relevant. The full annotation prompt is provided in Appendix C.3.

In the validation stage, GPT-4o attempted to answer the query using only the sentences marked as relevant, without access to the gold answer. If the model reproduced the correct answer, the annotations were retained; otherwise, the document was re-annotated using GPT-4o with an expanded 200-sentence window. These revised annotations were then processed with the same validation procedure. Data points failing both validations were discarded, as manual inspection indicated that they required reasoning over context spans too long for the models to reliably capture.

This process yielded 3,539 validated test instances from the original 5,735. Retrieval performance on this filtered set was evaluated using recall, precision, and nDCG. As shown in Table 5, SPScanner 1.3B consistently achieved the best performance across all metrics.

# 9 Conclusion

We introduce the Single-Pass Scanner, which excels in long document question answering comparing to RAG systems and approaches GPT-4o's performance for documents over 256k tokens despite being much smaller. Trained with our novel link-based synthetic data, the Single-Pass Scanner outperforms all state-of-the-art embedding models, such as NV-Embed-v2, while using fewer FLOPs and processing documents faster. By taking into account all prior document context, the model efficiently leverages long-range dependencies for answering questions about long and complex documents. Our approach eliminates the need for document chunking, a common limitation in current retrieval systems that often results in loss of context and reduced accuracy. Additionally, we propose a novel link-based synthetic data generation strategy that proves most effective for training, helping Single-Pass Scanners capture long-distance dependencies more effectively.

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

**Appendix: Table of Contents**

# A Dataset

## A.1 Test Sets & Validation Sets Overview

For testing, evaluation is done on 41 QA benchmark tasks, which are collected from the following long-document understanding benchmarks: Longbench (Bai et al., 2024), ∞ bench (Zhang et al., 2024b), L-eval (An et al., 2024a), LVeval (Yuan et al., 2024), Bamboo (Dong et al., 2024), ELITR bench (Thonet et al., 2024), docfinQA (Reddy et al., 2024), MuLD (Hudson & Al Moubayed, 2022). The tasks are freeform and multiple-choice questions on long documents, which range from 1,000 to 780,000 tokens.

Note, for clarity of presentation in the main paper, we categorize all data points from these 41 datasets into 4 categories based on the type of long documents reported by their original documentation. Each data point can belong to only one category, but within a dataset, data points can be distributed across multiple categories. Benchmark task statistics based on these 4 categories are provided in Table 6. Details of each dataset and the categories it belongs to can be found in Appendix A.2.

- Educational: wikipedia, college exams, English tests, etc.
- Creative: movie scripts, novels, screenplays, etc.
- Official: legal contract, financial documents, scientific papers, etc.
- Conversational: meeting transcripts, dialogues, etc.

We also record model performance averaged across all data points (not averaged by the 4 categories) in the "Average" column in tables. Model performance on each dataset is in Appendix A.4.

For validation sets, we take 100 data points each from the train sets of 8 benchmark tasks. We ensured and verified no document or question from our validation sets are in any test set. Validation sets and test sets are completely disjoint. We did not use the validation sets for training any models or for guiding any synthetic data generation. The sole purpose of our validation sets is for hyperparameter tuning of our Single-Pass Scanners and embedding models. See full details of validation sets in Appendix A.3.

| Category | Test Size (n) | Average Length (tokens) | Freeform Questions (n) | Multiple Choice Questions (n) | Number of Answers (n) | Average Answer Length (tokens) |
|---|---|---|---|---|---|---|
| Educational | 1967 | 45 159 | 1595 | 372 | 1.04 | 6.39 |
| Creative | 1733 | 120 585 | 1247 | 486 | 1.00 | 11.16 |
| Official | 1328 | 73 808 | 1156 | 172 | 1.11 | 23.18 |
| Conversational | 707 | 36 922 | 530 | 177 | 1.13 | 4.97 |
| Total | 5735 | 69 119 | 4528 | 1207 | 6061 | 11.67 |

Table 6: Document type statistics for the 41 benchmark tasks.

## A.2  Test Sets Statistics

| Dataset | Category | | | | Test Size | Average Length |
| --- | --- | --- | --- | --- | --- | --- |
| | Educational | Creative | Official | Conversational | (n) | (tokens) |
| narrativeqa | | ✓ | | | 200 | 30 551 |
| qasper | | | ✓ | | 200 | 5 039 |
| multifieldqa_en | ✓ | ✓ | ✓ | | 147 | 6 951 |
| hotpotqa | ✓ | | | | 200 | 12 802 |
| 2wikimqa | ✓ | | | | 199 | 7 152 |
| musique | ✓ | | | | 200 | 15 560 |
| longbook_choice_eng | | ✓ | | | 200 | 194 984 |
| longdialogue_qa_eng | | | | ✓ | 200 | 109 994 |
| longbook_qa_eng | | ✓ | | | 200 | 195 284 |
| loogle_CR_mixup_16k | ✓ | ✓ | ✓ | | 99 | 31 633 |
| loogle_CR_mixup_32k | ✓ | ✓ | ✓ | | 99 | 50 305 |
| loogle_CR_mixup_64k | ✓ | ✓ | ✓ | | 99 | 96 750 |
| loogle_CR_mixup_128k | ✓ | ✓ | ✓ | | 99 | 177 463 |
| loogle_CR_mixup_256k | ✓ | ✓ | ✓ | | 99 | 339 055 |
| loogle_MIR_mixup_16k | ✓ | ✓ | ✓ | | 139 | 33 240 |
| loogle_MIR_mixup_32k | ✓ | ✓ | ✓ | | 139 | 49 991 |
| loogle_MIR_mixup_64k | ✓ | ✓ | ✓ | | 139 | 97 818 |
| loogle_MIR_mixup_128k | ✓ | ✓ | ✓ | | 139 | 178 771 |
| loogle_MIR_mixup_256k | ✓ | ✓ | ✓ | | 139 | 340 596 |
| multifieldqa_en_mixup_16k | ✓ | ✓ | ✓ | | 101 | 28 194 |
| multifieldqa_en_mixup_32k | ✓ | ✓ | ✓ | | 101 | 52 810 |
| multifieldqa_en_mixup_64k | ✓ | ✓ | ✓ | | 101 | 101 375 |
| multifieldqa_en_mixup_128k | ✓ | ✓ | ✓ | | 101 | 197 624 |
| multifieldqa_en_mixup_256k | ✓ | ✓ | ✓ | | 101 | 390 300 |
| muld_CAC | | ✓ | | | 86 | 48 993 |
| ELITR_Bench_QA | | | | ✓ | 130 | 11 147 |
| altqa_4k | ✓ | | | | 199 | 3 223 |
| altqa_16k | ✓ | | | | 199 | 13 011 |
| meetingpred_4k | | | | ✓ | 100 | 3 676 |
| meetingpred_16k | | | | ✓ | 100 | 11 692 |
| meetingqa_4k | | | | ✓ | 86 | 2 731 |
| meetingqa_16k | | | | ✓ | 91 | 9 720 |
| paperqa_4k | | | ✓ | | 82 | 3 114 |
| paperqa_16k | | | ✓ | | 90 | 6 671 |
| tpo | ✓ | | | | 200 | 3 555 |
| financial_qa | | | ✓ | | 68 | 5 050 |
| legal_contract_qa | | | ✓ | | 130 | 25 529 |
| scientific_qa | | | ✓ | | 161 | 4 405 |
| quality | | ✓ | | | 200 | 5 959 |
| coursera | ✓ | | | | 172 | 8 269 |
| docfinQA | | | ✓ | | 200 | 212 751 |

Table 7: Dataset statistics for the 41 benchmark tasks. For larger benchmark tasks, we randomly sampled 200 data point instances from that task.

### A.3 Validation Sets Statistics

| Dataset | Test Size (n) | Average Length (tokens) |
|---|---|---|
| docfinQA | 100 | 142 328 |
| muld_CAC | 100 | 45 520 |
| ELITR_Bench | 100 | 10 328 |
| narrativeqa | 100 | 71 843 |
| qasper | 100 | 5 274 |
| wiki | 100 | 823 |
| hotpotqa | 100 | 1 313 |
| musique | 100 | 2 230 |

Table 8: Dataset statistics for validation set.

## A.4 GPT-4o as Generator

| Model | Average | narrativeqa | qasper_en | multifieldqa | hotpotqa | 2wikimqa | musique_eng | longbook_choice_eng | longdialogue_qa |
|---|---|---|---|---|---|---|---|---|---|
| | n = 5735 | n = 200 | n = 200 | n = 147 | n = 200 | n = 199 | n = 200 | n = 200 | n = 200 |
| | | | | **Retrieving 10 sentences** | | | | | |
| | | | | Accuracy | | | | | |
| SPScanner 1.3B | **54.1** | **44.5** | **48.5** | 81.0 | **72.5** | **74.9** | 57.5 | 60.0 | 21.0 |
| SPScanner 130M | 51.8 | 41.0 | 48.0 | **83.7** | 72.0 | 71.9 | 55.0 | 54.0 | **36.5** |
| BM25 | 37.4 | 21.5 | 28.0 | 66.0 | 69.0 | 41.7 | 42.5 | 44.0 | 9.5 |
| Contriever | 46.2 | 30.5 | 32.0 | 70.1 | 63.0 | 56.3 | 45.0 | 61.5 | 8.5 |
| Contriever-FT | 46.6 | 32.5 | 38.0 | 74.1 | 65.5 | 59.8 | 46.0 | 57.5 | 4.5 |
| GritLM | 49.4 | 32.5 | 43.5 | 81.0 | 69.5 | 64.3 | **58.0** | 66.5 | 7.5 |
| NV-Embed-v2-7B | 49.2 | 31.5 | 40.0 | 76.9 | 71.0 | 65.8 | 51.5 | **67.0** | 13.0 |
| Stella-1.5B | 46.8 | 33.0 | 36.0 | 76.2 | 70.0 | 61.8 | 54.5 | 60.5 | 9.0 |
| GTE-Qwen2-1.5B | 46.7 | 30.5 | 40.0 | 72.1 | 64.0 | 61.8 | 54.0 | 64.5 | 11.0 |
| | | | | F1 | | | | | |
| SPScanner 1.3B | **29.6** | **17.3** | 29.0 | 46.9 | **39.0** | **41.4** | 30.3 | N/A | 21.2 |
| SPScanner 130M | 28.6 | 17.1 | **30.3** | **48.1** | 36.7 | 36.1 | 30.7 | N/A | **37.3** |
| BM25 | 21.6 | 12.0 | 20.8 | 42.3 | 36.8 | 23.2 | 21.6 | N/A | 9.7 |
| Contriever | 25.0 | 13.0 | 22.8 | 41.7 | 30.4 | 30.9 | 24.1 | N/A | 9.5 |
| Contriever-FT | 25.5 | 14.0 | 24.6 | 45.2 | 32.6 | 32.4 | 26.2 | N/A | 5.4 |
| GritLM | 26.6 | 14.5 | 26.8 | 45.7 | 38.3 | 32.8 | 28.2 | N/A | 8.5 |
| NV-Embed-v2-7B | 27.1 | 16.1 | 25.7 | 45.5 | 36.1 | 35.2 | 28.0 | N/A | 12.6 |
| Stella-1.5B | 26.0 | 14.6 | 22.8 | 44.6 | 36.0 | 33.9 | 28.9 | N/A | 9.7 |
| GTE-Qwen2-1.5B | 25.8 | 14.0 | 25.0 | 44.1 | 31.8 | 31.7 | **31.7** | N/A | 10.6 |
| | | | | **Retrieving 50 sentences** | | | | | |
| | | | | Accuracy | | | | | |
| SPScanner 1.3B | **61.8** | **57.5** | **57.5** | 83.7 | **82.0** | **84.9** | **63.0** | 75.5 | 29.0 |
| SPScanner 130M | 60.0 | 50.0 | 53.5 | **87.1** | **82.0** | 78.9 | 61.0 | 64.0 | **37.0** |
| BM25 | 44.6 | 33.0 | 38.5 | 74.1 | 73.5 | 63.3 | 55.5 | 53.5 | 11.0 |
| Contriever | 53.1 | 37.5 | 48.5 | 79.6 | 75.5 | 73.9 | 58.5 | 67.5 | 16.0 |
| Contriever-FT | 54.8 | 44.5 | 49.5 | 82.3 | 77.0 | 82.4 | 57.5 | 69.5 | 14.5 |
| GritLM | 56.7 | 47.5 | 49.5 | 83.7 | 80.0 | 83.4 | 60.0 | 71.0 | 14.0 |
| NV-Embed-v2-7B | 56.6 | 46.0 | 52.5 | 83.0 | 77.0 | 82.4 | 61.0 | 70.5 | 12.0 |
| Stella-1.5B | 55.6 | 44.5 | 51.0 | 86.4 | 76.0 | 80.9 | 61.5 | 70.5 | 13.5 |
| GTE-Qwen2-1.5B | 54.6 | 46.0 | 49.5 | 83.0 | 75.5 | 77.9 | 58.5 | 74.0 | 15.5 |
| | | | | F1 | | | | | |
| SPScanner 1.3B | **32.2** | **20.1** | **32.6** | 46.6 | **48.2** | 46.7 | **37.1** | N/A | 28.4 |
| SPScanner 130M | 32.0 | 18.8 | 32.2 | **48.2** | 45.6 | 43.2 | 34.6 | N/A | **36.4** |
| BM25 | 24.8 | 15.8 | 24.8 | 44.9 | 37.5 | 35.5 | 29.0 | N/A | 10.6 |
| Contriever | 28.9 | 16.7 | 30.8 | 45.6 | 38.5 | 38.3 | 32.3 | N/A | 16.7 |
| Contriever-FT | 29.9 | 20.6 | 30.4 | 46.7 | 43.0 | 45.9 | 33.0 | N/A | 13.7 |
| GritLM | 29.9 | 19.6 | 30.3 | **48.2** | 39.9 | **49.1** | 34.0 | N/A | 12.5 |
| NV-Embed-v2-7B | 30.1 | 18.6 | 31.6 | 44.6 | 39.5 | 45.1 | 31.8 | N/A | 11.3 |
| Stella-1.5B | 29.6 | 17.8 | 30.1 | 47.8 | 42.5 | 43.1 | 34.7 | N/A | 12.9 |
| GTE-Qwen2-1.5B | 29.0 | 18.6 | 30.1 | 46.1 | 37.7 | 41.8 | 34.1 | N/A | 14.9 |
| | | | | **Retrieving 5 chunks** | | | | | |
| | | | | Accuracy | | | | | |
| BM25 | 49.1 | 39.5 | 48.0 | 75.5 | 76.5 | 70.4 | 53.5 | 60.5 | 9.5 |
| Contriever | 54.3 | 45.5 | 50.0 | 80.3 | 73.5 | 78.9 | 55.5 | 69.5 | 18.5 |
| Contriever-FT | 54.8 | **53.0** | 49.5 | 80.3 | 71.5 | 77.9 | 52.0 | 69.5 | 8.5 |
| GritLM | 57.2 | 46.5 | **52.5** | **85.0** | 76.5 | 81.4 | **58.5** | 69.0 | 30.0 |
| NV-Embed-v2-7B | **59.1** | 49.5 | 50.5 | **85.0** | **78.5** | **83.4** | 58.0 | 73.5 | **39.0** |
| Stella-1.5B | 56.8 | 43.5 | 48.5 | 83.0 | 75.5 | 81.9 | 52.0 | 74.0 | 22.5 |
| GTE-Qwen2-1.5B | 55.7 | 44.5 | **52.5** | 78.9 | 73.5 | 76.4 | 54.0 | **74.5** | 15.0 |
| M2-BERT | 38.7 | 29.0 | 42.0 | 66.0 | 67.5 | 60.8 | 48.5 | 46.5 | 7.5 |
| | | | | F1 | | | | | |
| BM25 | 26.0 | 17.4 | 29.3 | 42.9 | **41.7** | 39.5 | 30.4 | N/A | 9.2 |
| Contriever | 28.7 | 18.2 | 30.0 | 44.9 | 36.8 | 44.2 | 29.0 | N/A | 18.7 |
| Contriever-FT | 28.1 | 18.8 | 29.2 | **47.3** | 35.9 | 40.7 | 27.4 | N/A | 8.2 |
| GritLM | 29.9 | 18.9 | 29.7 | 46.1 | 39.5 | 45.2 | **31.4** | N/A | 29.8 |
| NV-Embed-v2-7B | **30.6** | **19.1** | 29.8 | 45.0 | 38.5 | 45.1 | 29.1 | N/A | **39.2** |
| Stella-1.5B | 29.9 | 19.0 | 29.1 | 46.6 | 40.0 | **45.8** | 27.8 | N/A | 21.0 |
| GTE-Qwen2-1.5B | 29.2 | **19.1** | **31.3** | 45.1 | 35.2 | 42.0 | 30.2 | N/A | 14.3 |
| M2-BERT | 21.4 | 14.6 | 25.6 | 38.7 | 35.1 | 31.2 | 23.6 | N/A | 7.8 |
| | | | | **Full context** | | | | | |
| | | | | Accuracy | | | | | |
| GPT-4o Full Context | 64.6 | 59.0 | 58.5 | 85.7 | 83.0 | 84.9 | 64.5 | 83.5 | 47.0 |
| | | | | F1 | | | | | |
| GPT-4o Full Context | 33.0 | 22.0 | 33.8 | 48.7 | 50.2 | 46.8 | 39.8 | N/A | 40.0 |

Table 9: QA accuracy across 41 datasets with GPT-4o as generator. When not paired with a retriever, GPT-4o is provided with the full document in-context. Results continue to next page.

| Model | longbook_qa_eng n = 200 | loogle_CR_mixup_16k n = 99 | loogle_CR_mixup_32k n = 99 | loogle_CR_mixup_64k n = 99 | loogle_CR_mixup_128k n = 99 | loogle_MIR_mixup_16k n = 139 | loogle_CR_mixup_256k n = 99 | loogle_MIR_mixup_32k n = 139 |
|---|---|---|---|---|---|---|---|---|
| | | | | **Retrieving 10 sentences** | | | | |
| | | | | Accuracy | | | | |
| SPScanner 1.3B | **41.0** | 36.4 | 37.4 | **39.4** | **40.4** | **32.4** | **41.4** | **36.0** |
| SPScanner 130M | 37.5 | **39.4** | 37.4 | 37.4 | 32.3 | 25.9 | 40.4 | 24.5 |
| BM25 | 16.5 | 19.2 | 18.2 | 18.2 | 16.2 | 9.4 | 16.2 | 7.9 |
| Contriever | 24.5 | 32.3 | 34.3 | 30.3 | 31.3 | 27.3 | 27.3 | 26.6 |
| Contriever-FT | 26.5 | 32.3 | 31.3 | 25.3 | 30.3 | 28.1 | 28.3 | 28.1 |
| GritLM | 28.5 | 35.4 | **38.4** | 37.4 | 32.3 | 29.5 | 32.3 | 26.6 |
| NV-Embed-v2-7B | 30.5 | 35.4 | 33.3 | 31.3 | 30.3 | 30.9 | 30.3 | 30.2 |
| Stella-1.5B | 24.5 | 33.3 | 30.3 | 33.3 | 30.3 | 25.9 | 29.3 | 24.5 |
| GTE-Qwen2-1.5B | 24.5 | 32.3 | 35.4 | 37.4 | 37.4 | 21.6 | 38.4 | 20.1 |
| | | | | F1 | | | | |
| SPScanner 1.3B | 16.5 | **22.4** | **21.3** | **23.6** | **23.0** | **25.6** | **24.3** | **25.3** |
| SPScanner 130M | **18.2** | 21.0 | 21.2 | 21.6 | 19.0 | 23.5 | 20.7 | 22.3 |
| BM25 | 9.5 | 18.2 | 17.9 | 17.2 | 16.3 | 15.2 | 17.3 | 14.4 |
| Contriever | 10.8 | 17.8 | 18.5 | 19.8 | 18.8 | 21.1 | 18.7 | 22.4 |
| Contriever-FT | 12.6 | 19.2 | 19.1 | 18.4 | 19.3 | 21.3 | 17.0 | 21.5 |
| GritLM | 14.4 | 20.3 | 20.6 | 22.2 | 20.8 | 21.8 | 20.1 | 22.3 |
| NV-Embed-v2-7B | 13.9 | 19.7 | 20.4 | 20.5 | 19.4 | 23.6 | 19.5 | 23.4 |
| Stella-1.5B | 14.2 | 20.1 | 18.9 | 20.0 | 19.9 | 21.1 | 20.4 | 20.1 |
| GTE-Qwen2-1.5B | 12.9 | 20.1 | 20.6 | 20.9 | 20.6 | 19.1 | 20.7 | 19.3 |
| | | | | **Retrieving 50 sentences** | | | | |
| | | | | Accuracy | | | | |
| SPScanner 1.3B | **54.0** | **56.6** | 51.5 | 46.5 | 51.5 | 43.2 | **48.5** | 41.0 |
| SPScanner 130M | 50.0 | 55.6 | 51.5 | 46.5 | 47.5 | **45.3** | 47.5 | **46.0** |
| BM25 | 26.5 | 31.3 | 25.3 | 22.2 | 26.3 | 16.5 | 25.3 | 14.4 |
| Contriever | 33.5 | 40.4 | 35.4 | 40.4 | 38.4 | 36.0 | 41.4 | 30.9 |
| Contriever-FT | 39.0 | 41.4 | 39.4 | 35.4 | 39.4 | 34.5 | 34.3 | 34.5 |
| GritLM | 40.5 | 46.5 | 43.4 | **46.5** | 39.4 | 33.8 | 44.4 | 36.7 |
| NV-Embed-v2-7B | 42.0 | 51.5 | 46.5 | **46.5** | 51.5 | 35.3 | 44.4 | 38.1 |
| Stella-1.5B | 37.0 | 45.5 | 44.4 | 44.4 | 47.5 | 33.8 | 40.4 | 35.3 |
| GTE-Qwen2-1.5B | 37.0 | 46.5 | 39.4 | 37.4 | 39.4 | 33.8 | 35.4 | 33.1 |
| | | | | F1 | | | | |
| SPScanner 1.3B | **23.7** | 24.3 | 23.7 | **24.8** | 24.7 | **26.2** | 23.1 | **29.0** |
| SPScanner 130M | 23.1 | **25.1** | **25.5** | 24.6 | 24.2 | 25.1 | 24.3 | 27.9 |
| BM25 | 13.0 | 19.6 | 17.8 | 16.4 | 19.8 | 17.6 | 17.2 | 15.4 |
| Contriever | 15.7 | 23.0 | 23.1 | 23.3 | 22.7 | 22.4 | 20.1 | 21.7 |
| Contriever-FT | 18.0 | 25.1 | 21.7 | 22.1 | 23.4 | 25.7 | 21.7 | 25.4 |
| NV-Embed-v2-7B | 18.3 | 23.4 | 23.8 | 22.8 | **25.0** | 25.2 | 22.2 | 27.1 |
| Stella-1.5B | 18.5 | 24.6 | 21.5 | 23.4 | 23.6 | 24.5 | **23.8** | 24.0 |
| GTE-Qwen2-1.5B | 15.9 | 24.4 | 24.9 | 22.5 | 19.9 | 21.9 | 20.9 | 22.4 |
| | | | | **Retrieving 5 chunks** | | | | |
| | | | | Accuracy | | | | |
| BM25 | 26.5 | 36.4 | 44.4 | 35.4 | 35.4 | 18.7 | 35.4 | 18.0 |
| Contriever | 40.0 | 48.5 | 51.5 | 42.4 | 43.4 | 24.5 | 41.4 | 23.0 |
| Contriever-FT | 40.5 | 49.5 | 43.4 | 41.4 | 39.4 | 30.2 | 45.5 | 28.8 |
| GritLM | 48.5 | 52.5 | 51.5 | 45.5 | 48.5 | 31.7 | 46.5 | **36.7** |
| NV-Embed-v2-7B | **49.5** | 52.5 | **52.5** | 56.6 | 58.6 | 30.9 | **50.5** | 34.5 |
| Stella-1.5B | 46.5 | **54.5** | 47.5 | 49.5 | 50.5 | **34.5** | 44.4 | 31.7 |
| GTE-Qwen2-1.5B | 41.5 | 49.5 | 49.5 | 41.4 | 42.4 | 29.5 | 39.4 | 29.5 |
| M2-BERT | 14.5 | 25.3 | 19.2 | 25.3 | 23.2 | 9.4 | 16.2 | 12.9 |
| | | | | F1 | | | | |
| BM25 | 12.7 | 21.0 | 21.1 | 19.6 | 20.2 | 17.4 | 19.2 | 14.8 |
| Contriever | 18.6 | 22.4 | 23.3 | 23.5 | 21.2 | 19.4 | 20.6 | 21.3 |
| Contriever-FT | 19.4 | 21.0 | 20.5 | 21.0 | 21.3 | 22.4 | 20.3 | 19.1 |
| GritLM | 20.1 | 21.9 | 22.1 | 22.7 | 20.4 | 22.5 | 21.4 | 22.5 |
| NV-Embed-v2-7B | **20.6** | 22.0 | 22.4 | 21.3 | 22.9 | **23.3** | **22.9** | **23.1** |
| Stella-1.5B | 20.5 | 24.0 | **24.6** | **24.2** | **23.1** | 21.1 | 21.7 | 21.8 |
| GTE-Qwen2-1.5B | 17.9 | **24.1** | 23.8 | 20.7 | 21.6 | 22.0 | 21.7 | 21.6 |
| M2-BERT | 8.2 | 18.8 | 17.4 | 16.4 | 15.5 | 14.8 | 15.0 | 14.5 |
| | | | | **Full context** | | | | |
| | | | | Accuracy | | | | |
| GPT-4o Full Context | 66.5 | 51.5 | 59.6 | 48.5 | 57.6 | 48.2 | 49.5 | 46.0 |
| | | | | F1 | | | | |
| GPT-4o Full Context | 19.8 | 26.9 | 27.2 | 22.3 | 19.2 | 29.8 | 13.1 | 27.8 |

Table 10: Results continued

| Model | loogle_MIR_mixup_64k n = 139 | loogle_MIR_mixup_128k n = 139 | loogle_MIR_mixup_256k n = 139 | multifieldqa_en_mixup_16k n = 101 | multifieldqa_en_mixup_32k n = 101 | multifieldqa_en_mixup_64k n = 101 | multifieldqa_en_mixup_128k n = 101 |
|---|---|---|---|---|---|---|---|
| **Retrieving 10 sentences** | | | | | | | |
| *Accuracy* | | | | | | | |
| SPScanner 1.3B | **33.1** | **34.5** | **32.4** | **52.5** | 55.4 | **53.5** | 52.5 |
| SPScanner 130M | 25.9 | 25.2 | 23.7 | 51.5 | **56.4** | 52.5 | **53.5** |
| BM25 | 8.6 | 5.0 | 5.8 | 36.6 | 38.6 | 38.6 | 34.7 |
| Contriever | 25.2 | 27.3 | 25.9 | 47.5 | 45.5 | 46.5 | 45.5 |
| Contriever-FT | 25.2 | 23.0 | 22.3 | 48.5 | 51.5 | 47.5 | 43.6 |
| GritLM | 28.8 | 30.2 | 27.3 | 46.5 | 45.5 | 42.6 | 40.6 |
| NV-Embed-v2-7B | 28.1 | 29.5 | 29.5 | 47.5 | 48.5 | 46.5 | 47.5 |
| Stella-1.5B | 23.7 | 24.5 | 25.2 | 45.5 | 41.6 | 48.5 | 43.6 |
| GTE-Qwen2-1.5B | 20.1 | 20.1 | 19.4 | 45.5 | 38.6 | 36.6 | 42.6 |
| *F1* | | | | | | | |
| SPScanner 1.3B | **25.2** | **25.7** | **25.1** | **28.7** | 29.7 | **29.4** | 29.0 |
| SPScanner 130M | 23.3 | 22.2 | 22.1 | 26.9 | **31.1** | 29.3 | **30.1** |
| BM25 | 14.1 | 12.0 | 12.4 | 26.1 | 23.6 | 22.9 | 24.9 |
| Contriever | 19.9 | 21.6 | 20.4 | 26.5 | 26.3 | 26.8 | 27.9 |
| Contriever-FT | 23.3 | 21.7 | 18.9 | 27.7 | 28.1 | 26.5 | 27.5 |
| GritLM | 23.3 | 23.5 | 22.4 | 25.7 | 26.9 | 27.1 | 26.2 |
| NV-Embed-v2-7B | 22.3 | 21.5 | 23.6 | 26.9 | 27.3 | 28.1 | 26.2 |
| Stella-1.5B | 21.5 | 21.5 | 22.0 | 26.0 | 26.8 | 27.0 | 26.2 |
| GTE-Qwen2-1.5B | 19.9 | 19.0 | 19.5 | 27.8 | 26.4 | 25.9 | 27.0 |
| **Retrieving 50 sentences** | | | | | | | |
| *Accuracy* | | | | | | | |
| SPScanner 1.3B | 43.9 | **42.4** | 41.0 | **55.4** | 59.4 | 49.5 | 52.5 |
| SPScanner 130M | 39.6 | 39.6 | 37.4 | 54.5 | 59.4 | 54.5 | 58.4 |
| BM25 | 15.8 | 12.2 | 11.5 | 42.6 | 40.6 | 43.6 | 45.5 |
| Contriever | **73.9** | 25.0 | **44.6** | 20.0 | **87.2** | **78.0** | **81.7** |
| Contriever-FT | 30.9 | 33.8 | 25.9 | 49.5 | 57.4 | 48.5 | 54.5 |
| GritLM | 35.3 | 32.4 | 37.4 | 51.5 | 50.5 | 47.5 | 47.5 |
| NV-Embed-v2-7B | 32.4 | 36.7 | 31.7 | 48.5 | 51.5 | 46.5 | 49.5 |
| Stella-1.5B | 34.5 | 33.8 | 29.5 | 53.5 | 56.4 | 48.5 | 51.5 |
| GTE-Qwen2-1.5B | 30.2 | 33.1 | 32.4 | 47.5 | 52.5 | 53.5 | 47.5 |
| *F1* | | | | | | | |
| SPScanner 1.3B | **26.5** | 25.7 | **26.4** | 28.6 | 28.8 | 27.2 | 28.2 |
| SPScanner 130M | 26.2 | 26.4 | 26.2 | **30.8** | **30.7** | **29.6** | **30.9** |
| BM25 | 16.0 | 15.4 | 13.9 | 27.4 | 25.9 | 25.6 | 26.6 |
| Contriever | 22.8 | 22.0 | 22.8 | 29.1 | 27.2 | 26.9 | 27.8 |
| Contriever-FT | 25.4 | 23.8 | 22.2 | 28.0 | 29.1 | 27.2 | 28.5 |
| GritLM | 23.3 | 23.9 | 24.2 | 27.9 | 27.6 | 28.9 | 27.2 |
| NV-Embed-v2-7B | 25.6 | **26.6** | 25.4 | 28.5 | 27.1 | 27.0 | 26.8 |
| Stella-1.5B | 23.7 | 25.5 | 24.2 | 28.0 | 28.1 | 28.2 | 28.6 |
| GTE-Qwen2-1.5B | 22.1 | 24.1 | 23.8 | 29.2 | 26.3 | 28.3 | 27.1 |
| **Retrieving 5 chunks** | | | | | | | |
| *Accuracy* | | | | | | | |
| BM25 | 12.2 | 15.8 | 13.7 | 44.6 | 52.5 | 54.5 | 53.5 |
| Contriever | 25.9 | 24.5 | 25.2 | 50.5 | **53.5** | 57.4 | 55.4 |
| Contriever-FT | 29.5 | 23.0 | 25.9 | 51.5 | 50.5 | 51.5 | 47.5 |
| GritLM | 27.3 | 30.9 | 27.3 | 45.5 | 51.5 | **62.4** | 53.5 |
| NV-Embed-v2-7B | 31.7 | **35.3** | **30.9** | **52.5** | 50.5 | 49.5 | 50.5 |
| Stella-1.5B | 32.4 | 28.1 | **30.9** | 48.5 | 52.5 | 59.4 | 51.5 |
| GTE-Qwen2-1.5B | **34.5** | 28.1 | 28.8 | 51.5 | 51.5 | 53.5 | **58.4** |
| M2-BERT | 6.5 | 4.3 | 5.8 | 25.7 | 24.8 | 15.8 | 14.9 |
| *F1* | | | | | | | |
| BM25 | 15.9 | 16.1 | 14.5 | 27.6 | 26.8 | 28.1 | 29.6 |
| Contriever | 21.4 | 20.5 | 22.1 | 28.1 | 29.4 | 29.0 | 29.1 |
| Contriever-FT | 21.9 | 19.0 | 21.3 | 28.9 | 27.5 | 28.1 | 26.6 |
| GritLM | 20.7 | **23.7** | 21.8 | 26.7 | 28.7 | 28.4 | **30.0** |
| NV-Embed-v2-7B | 22.2 | 23.2 | 20.5 | 29.9 | 27.5 | 27.9 | 28.6 |
| Stella-1.5B | **23.0** | 22.4 | **22.3** | 27.5 | 28.3 | **31.1** | 29.6 |
| GTE-Qwen2-1.5B | 21.4 | 22.8 | 20.6 | **30.1** | **29.6** | 29.7 | 28.9 |
| M2-BERT | 12.4 | 9.9 | 10.8 | 18.1 | 18.5 | 16.5 | 15.9 |
| **Full context** | | | | | | | |
| *Accuracy* | | | | | | | |
| GPT-4o Full Context | 45.3 | 41.0 | 36.7 | 57.4 | 62.4 | 55.4 | 56.4 |
| *F1* | | | | | | | |
| GPT-4o Full Context | 27.7 | 20.5 | 13.3 | 32.2 | 32.3 | 28.8 | 24.9 |

Table 11: Results continued

| Model | altqa _4k n = 199 | altqa _16k n = 199 | meetingpred _4k n = 100 | multifieldqa_en _mixup_256k n = 101 | meetingpred _16k n = 100 | meetingqa _4k n = 86 | meetingqa _16k n = 91 | paperqa _4k n = 82 | paperqa _16k n = 90 |
|---|---|---|---|---|---|---|---|---|---|
| | | | | **Retrieving 10 sentences** | | | | | |
| | | | | Accuracy | | | | | |
| SPScanner 1.3B | **77.4** | 71.4 | 26.0 | **58.4** | **21.0** | 80.2 | 76.9 | **84.1** | 81.1 |
| SPScanner 130M | 73.9 | 69.8 | 18.0 | 49.5 | 19.0 | 80.2 | 74.7 | 80.5 | 77.8 |
| BM25 | 66.3 | 55.3 | 14.0 | 32.7 | 14.0 | 84.9 | 72.5 | 82.9 | **83.3** |
| Contriever | 73.9 | 73.9 | 25.0 | 44.6 | 20.0 | **87.2** | 78.0 | 81.7 | 78.9 |
| Contriever-FT | 74.9 | 69.8 | 19.0 | 45.5 | 20.0 | 79.1 | **80.2** | 78.0 | 81.1 |
| GritLM | **77.4** | 70.4 | 26.0 | 45.5 | 19.0 | 82.6 | 78.0 | 82.9 | 81.1 |
| NV-Embed-v2-7B | 75.4 | 71.9 | 30.0 | 45.5 | 20.0 | 81.4 | 75.8 | 81.7 | 80.0 |
| Stella-1.5B | 70.9 | 73.4 | **34.0** | 43.6 | 14.0 | 77.9 | 74.7 | 79.3 | 80.0 |
| GTE-Qwen2-1.5B | 72.4 | **74.4** | 28.0 | 37.6 | 18.0 | 79.1 | 79.1 | 78.0 | 80.0 |
| | | | | F1 | | | | | |
| SPScanner 1.3B | **77.4** | 71.4 | 26.3 | 28.4 | **19.9** | N/A | N/A | N/A | N/A |
| SPScanner 130M | 73.9 | 69.8 | 17.0 | **29.5** | 17.7 | N/A | N/A | N/A | N/A |
| BM25 | 66.3 | 55.3 | 11.8 | 24.8 | 13.0 | N/A | N/A | N/A | N/A |
| Contriever | 73.9 | 73.9 | 22.1 | 26.6 | 17.7 | N/A | N/A | N/A | N/A |
| Contriever-FT | 74.9 | 69.8 | 16.4 | 27.6 | 18.1 | N/A | N/A | N/A | N/A |
| GritLM | **77.4** | 70.4 | 23.4 | 26.8 | 17.1 | N/A | N/A | N/A | N/A |
| NV-Embed-v2-7B | 75.4 | 71.9 | 28.0 | 28.0 | 16.8 | N/A | N/A | N/A | N/A |
| Stella-1.5B | 70.9 | 73.4 | 29.4 | 25.8 | 12.2 | N/A | N/A | N/A | N/A |
| GTE-Qwen2-1.5B | 72.4 | **74.4** | 25.6 | 24.9 | 16.9 | N/A | N/A | N/A | N/A |
| | | | | **Retrieving 50 sentences** | | | | | |
| | | | | Accuracy | | | | | |
| SPScanner 1.3B | 81.9 | **79.9** | 37.0 | 50.5 | **35.0** | 84.9 | **84.6** | 85.4 | 85.6 |
| SPScanner 130M | **83.9** | 77.9 | 28.0 | **55.4** | 24.0 | 82.6 | 78.0 | 80.5 | 85.6 |
| BM25 | 72.9 | 67.8 | 29.0 | 40.6 | 14.0 | 77.9 | 79.1 | 81.7 | 82.2 |
| Contriever | 83.4 | 75.9 | 30.0 | 48.5 | 26.0 | 84.9 | 80.2 | 80.5 | 84.4 |
| Contriever-FT | 83.4 | 73.9 | 37.0 | 47.5 | 21.0 | 82.6 | 83.5 | 84.1 | 85.6 |
| GritLM | 81.4 | 74.4 | 29.0 | 47.5 | 27.0 | **88.4** | 80.2 | **86.6** | **86.7** |
| NV-Embed-v2-7B | 80.4 | 76.9 | **39.0** | 50.5 | 27.0 | 84.9 | 80.2 | 79.3 | **86.7** |
| Stella-1.5B | 79.9 | 75.4 | 31.0 | 53.5 | 23.0 | 81.4 | 75.8 | **86.6** | 82.2 |
| GTE-Qwen2-1.5B | 81.4 | 73.9 | 33.0 | 51.5 | 21.0 | 80.2 | 80.2 | 81.7 | 85.6 |
| | | | | F1 | | | | | |
| SPScanner 1.3B | 81.9 | **79.9** | **33.0** | 27.0 | **31.7** | N/A | N/A | N/A | N/A |
| SPScanner 130M | **83.9** | 77.9 | 24.4 | **30.7** | 22.2 | N/A | N/A | N/A | N/A |
| BM25 | 72.9 | 67.8 | 24.1 | 26.0 | 12.6 | N/A | N/A | N/A | N/A |
| Contriever | 83.4 | 75.9 | 26.6 | 27.9 | 23.9 | N/A | N/A | N/A | N/A |
| Contriever-FT | 83.4 | 73.9 | 31.7 | 28.7 | 19.1 | N/A | N/A | N/A | N/A |
| GritLM | 81.4 | 74.4 | 24.9 | 26.8 | 24.5 | N/A | N/A | N/A | N/A |
| NV-Embed-v2-7B | 80.4 | 76.9 | 35.5 | 28.0 | 24.5 | N/A | N/A | N/A | N/A |
| Stella-1.5B | 79.9 | 75.4 | 28.1 | 27.9 | 19.6 | N/A | N/A | N/A | N/A |
| GTE-Qwen2-1.5B | 81.4 | 73.9 | 29.8 | 27.0 | 18.9 | N/A | N/A | N/A | N/A |
| | | | | **Retrieving 5 chunks** | | | | | |
| | | | | Accuracy | | | | | |
| BM25 | 69.8 | 60.8 | 33.0 | 50.5 | 18.0 | 81.4 | **84.6** | 79.3 | 82.2 |
| Contriever | 79.4 | 72.4 | 34.0 | 50.5 | 21.0 | **83.7** | 79.1 | 80.5 | 84.4 |
| Contriever-FT | 80.9 | 70.9 | 36.0 | **56.4** | 20.0 | 80.2 | 81.3 | 79.3 | **88.9** |
| GritLM | 80.9 | 74.9 | 39.0 | 49.5 | 23.0 | 82.6 | 78.0 | **82.9** | 81.1 |
| NV-Embed-v2-7B | 81.9 | **79.4** | 40.0 | 52.5 | **27.0** | 83.7 | 81.3 | 80.5 | 84.4 |
| Stella-1.5B | 80.9 | 72.9 | **42.0** | 50.5 | 24.0 | 80.2 | 81.3 | 80.5 | 83.3 |
| GTE-Qwen2-1.5B | **82.4** | 75.4 | 31.0 | 51.5 | 25.0 | 82.6 | 81.3 | 81.7 | 83.3 |
| M2-BERT | 68.8 | 57.3 | 34.0 | 10.9 | 16.0 | 83.7 | 74.7 | 78.0 | 87.8 |
| | | | | F1 | | | | | |
| BM25 | 69.3 | 60.8 | 27.5 | 27.6 | 16.9 | N/A | N/A | N/A | N/A |
| Contriever | 79.4 | 72.4 | 29.9 | 29.4 | 18.6 | N/A | N/A | N/A | N/A |
| Contriever-FT | 80.9 | 70.9 | 31.4 | 29.0 | 17.2 | N/A | N/A | N/A | N/A |
| GritLM | 80.9 | 74.9 | 33.8 | 29.3 | 20.2 | N/A | N/A | N/A | N/A |
| NV-Embed-v2-7B | 81.9 | **79.4** | **34.6** | 28.7 | **24.7** | N/A | N/A | N/A | N/A |
| Stella-1.5B | 80.9 | 72.9 | 35.8 | **29.9** | 21.1 | N/A | N/A | N/A | N/A |
| GTE-Qwen2-1.5B | **82.4** | 75.4 | 27.9 | 29.0 | 23.2 | N/A | N/A | N/A | N/A |
| M2-BERT | 68.9 | 57.3 | 29.2 | 14.6 | 13.2 | N/A | N/A | N/A | N/A |
| | | | | **Full context** | | | | | |
| | | | | Accuracy | | | | | |
| GPT-4o Full Context | 81.9 | 77.4 | 57.0 | 51.5 | 52.0 | 83.7 | 75.8 | 84.1 | 83.3 |
| | | | | F1 | | | | | |
| GPT-4o Full Context | 81.9 | 77.4 | 50.0 | 21.5 | 46.1 | N/A | N/A | N/A | N/A |

Table 12: Results continued

| Model | tpo | financial _qa | legal_contract _qa | scientific _qa | quality | coursera | docfinQA | muld _CAC | ELITR _Bench |
|---|---|---|---|---|---|---|---|---|---|
| | n = 200 | n = 68 | n = 130 | n = 161 | n = 200 | n = 172 | n = 200 | n = 86 | n = 130 |
| **Retrieving 10 sentences** | | | | | | | | | |
| *Accuracy* | | | | | | | | | |
| SPScanner 1.3B | 81.0 | 66.2 | **64.6** | **53.4** | 65.5 | 73.8 | **32.5** | 76.7 | **49.2** |
| SPScanner 130M | 80.0 | **67.6** | 59.2 | **53.4** | 61.5 | 74.4 | 29.5 | 84.9 | 46.2 |
| BM25 | 83.5 | 38.2 | 31.5 | 32.9 | 56.5 | 73.3 | 0.5 | 72.1 | 27.7 |
| Contriever | 87.5 | 30.9 | 52.3 | 37.9 | 61.0 | 79.1 | 7.5 | 81.4 | 29.2 |
| Contriever-FT | 76.0 | 41.2 | 61.5 | 44.7 | **68.0** | 74.4 | 11.0 | 79.1 | 36.9 |
| GritLM | **88.0** | 48.5 | 58.5 | 40.4 | 62.5 | **80.2** | 20.5 | 83.7 | 34.6 |
| NV-Embed-v2-7B | 79.5 | 57.4 | 56.9 | 43.5 | 62.5 | 73.8 | 17.5 | 84.9 | 35.4 |
| Stella-1.5B | 80.0 | 41.2 | 60.8 | 42.2 | 62.5 | 72.1 | 17.5 | 80.2 | 31.5 |
| GTE-Qwen2-1.5B | 78.5 | 52.9 | 53.1 | 41.0 | 66.5 | 72.1 | 15.0 | **86.0** | 33.1 |
| *F1* | | | | | | | | | |
| SPScanner 1.3B | N/A | **43.2** | **24.2** | **28.7** | N/A | N/A | 2.1 | N/A | **24.8** |
| SPScanner 130M | N/A | 42.5 | **24.2** | 27.4 | N/A | N/A | **2.4** | N/A | 23.1 |
| BM25 | N/A | 34.6 | 19.9 | 18.1 | N/A | N/A | 0.5 | N/A | 18.0 |
| Contriever | N/A | 35.9 | 22.6 | 20.0 | N/A | N/A | 1.0 | N/A | 18.0 |
| Contriever-FT | N/A | 35.1 | 24.7 | 24.2 | N/A | N/A | 1.6 | N/A | 20.2 |
| GritLM | N/A | 36.1 | 22.8 | 22.4 | N/A | N/A | 2.1 | N/A | 20.3 |
| NV-Embed-v2-7B | N/A | 40.2 | 23.6 | 23.8 | N/A | N/A | 2.0 | N/A | 21.4 |
| Stella-1.5B | N/A | 36.5 | 22.9 | 23.5 | N/A | N/A | 2.3 | N/A | 19.7 |
| GTE-Qwen2-1.5B | N/A | 35.7 | 23.5 | 22.8 | N/A | N/A | 1.5 | N/A | 19.9 |
| **Retrieving 50 sentences** | | | | | | | | | |
| *Accuracy* | | | | | | | | | |
| SPScanner 1.3B | 88.0 | 76.5 | **63.8** | 59.0 | **73.0** | 79.1 | **48.5** | **90.7** | 59.2 |
| SPScanner 130M | 80.0 | **79.4** | 53.8 | **59.6** | 71.0 | 73.8 | 47.5 | 89.5 | **63.1** |
| BM25 | 79.0 | 63.2 | 43.8 | 41.6 | 64.0 | 73.3 | 1.5 | 80.2 | 33.8 |
| Contriever | 81.5 | 67.6 | 56.2 | 50.9 | 70.5 | 72.7 | 13.0 | 86.0 | 47.7 |
| Contriever-FT | 80.0 | 69.1 | 57.7 | 54.0 | **73.0** | 72.1 | 20.5 | 84.9 | 60.8 |
| GritLM | **89.5** | 72.1 | 59.2 | 54.0 | 71.0 | **82.6** | 30.0 | 87.2 | 55.4 |
| NV-Embed-v2-7B | 80.0 | 66.2 | 61.5 | 54.7 | 70.5 | 76.2 | 27.5 | 88.4 | 55.4 |
| Stella-1.5B | 78.0 | 73.5 | 60.0 | 50.9 | 71.5 | 72.1 | 29.0 | 88.4 | 45.4 |
| GTE-Qwen2-1.5B | 79.0 | 67.6 | 57.7 | 57.1 | 71.0 | 72.1 | 28.0 | 88.4 | 52.3 |
| *F1* | | | | | | | | | |
| SPScanner 1.3B | N/A | **41.6** | 24.5 | **30.1** | N/A | N/A | **4.0** | N/A | 27.1 |
| SPScanner 130M | N/A | 41.3 | 23.3 | 29.1 | N/A | N/A | 3.0 | N/A | **28.4** |
| BM25 | N/A | 37.3 | 22.1 | 23.1 | N/A | N/A | 0.5 | N/A | 20.4 |
| Contriever | N/A | 39.6 | **25.2** | 27.2 | N/A | N/A | 1.3 | N/A | 23.8 |
| Contriever-FT | N/A | 40.2 | 23.8 | 27.5 | N/A | N/A | 2.0 | N/A | 25.9 |
| GritLM | N/A | 40.0 | 24.2 | 29.4 | N/A | N/A | 2.9 | N/A | 25.4 |
| NV-Embed-v2-7B | N/A | 40.0 | 24.1 | 29.3 | N/A | N/A | 2.9 | N/A | 25.4 |
| Stella-1.5B | N/A | 40.4 | 24.2 | 27.0 | N/A | N/A | 2.7 | N/A | 23.9 |
| GTE-Qwen2-1.5B | N/A | 39.7 | 24.1 | 29.8 | N/A | N/A | 3.5 | N/A | 24.7 |
| **Retrieving 5 chunks** | | | | | | | | | |
| *Accuracy* | | | | | | | | | |
| BM25 | 79.5 | 60.3 | 37.7 | 52.2 | 70.0 | **73.8** | 4.5 | 86.0 | 58.5 |
| Contriever | 77.0 | 66.2 | 39.2 | 53.4 | 70.5 | 73.3 | 35.5 | 86.0 | 63.1 |
| Contriever-FT | 79.5 | 69.1 | **61.5** | 55.3 | 74.0 | 71.5 | 37.5 | 87.2 | 57.7 |
| GritLM | 79.5 | **77.9** | 51.5 | 55.3 | 74.0 | **73.8** | 36.5 | 84.9 | 61.5 |
| NV-Embed-v2-7B | 79.0 | 70.6 | 56.2 | 53.4 | **74.5** | 72.7 | 44.5 | **90.7** | **65.4** |
| Stella-1.5B | 79.0 | 69.1 | 47.7 | 53.4 | 73.0 | 73.3 | 41.0 | 86.0 | **65.4** |
| GTE-Qwen2-1.5B | **80.0** | 69.1 | 46.9 | **57.1** | 69.5 | **73.8** | **46.0** | 86.0 | 63.1 |
| M2-BERT | 77.0 | 48.5 | 26.9 | 50.9 | 63.5 | 70.3 | 11.5 | 82.6 | 32.3 |
| *F1* | | | | | | | | | |
| BM25 | N/A | 40.5 | 21.7 | 26.4 | N/A | N/A | 1.1 | N/A | 26.5 |
| Contriever | N/A | 37.5 | 21.3 | 26.5 | N/A | N/A | 3.5 | N/A | 27.9 |
| Contriever-FT | N/A | 40.4 | 23.5 | 28.2 | N/A | N/A | 3.1 | N/A | 27.7 |
| GritLM | N/A | 41.7 | 23.0 | 27.2 | N/A | N/A | 3.0 | N/A | 27.8 |
| NV-Embed-v2-7B | N/A | 41.7 | **24.0** | 28.1 | N/A | N/A | 3.2 | N/A | 27.8 |
| Stella-1.5B | N/A | **42.1** | 22.4 | 26.7 | N/A | N/A | 3.4 | N/A | 28.3 |
| GTE-Qwen2-1.5B | N/A | 40.6 | 22.3 | **29.0** | N/A | N/A | **3.7** | N/A | **28.4** |
| M2-BERT | N/A | 37.4 | 19.0 | 25.0 | N/A | N/A | 0.7 | N/A | 19.5 |
| **Full context** | | | | | | | | | |
| *Accuracy* | | | | | | | | | |
| GPT-4o Full Context | 84.0 | 67.6 | 63.1 | 62.7 | 83.5 | 73.8 | 51.5 | 94.2 | 73.8 |
| *F1* | | | | | | | | | |
| GPT-4o Full Context | N/A | 43.1 | 24.2 | 30.0 | N/A | N/A | 4.4 | N/A | 31.6 |

Table 13: Results continued

## A.5 Llama-3.1 as Generator

| Model | Average | narrativeqa | qasper_en | multifieldqa | hotpotqa | 2wikimqa | musique_choice_eng | longbook_eng | longdialogue_qa |
|---|---|---|---|---|---|---|---|---|---|
| | n = 5735 | n = 200 | n = 200 | n = 147 | n = 200 | n = 199 | n = 200 | n = 200 | n = 200 |
| **Llama-3.1 8B** | | | | | | | | | |
| *Accuracy* | | | | | | | | | |
| SPScanner 1.3B | **47.9** | 44.5 | 50.5 | **86.4** | 64.5 | 48.2 | 36.0 | **49.5** | 13.0 |
| SPScanner 130M | 47.4 | 40.0 | **51.5** | 85.7 | **68.0** | 49.7 | **42.0** | 49.0 | **19.0** |
| GritLM | 36.5 | **44.7** | 47.0 | 84.4 | 67.0 | **51.8** | 37.5 | 47.0 | 9.5 |
| BM25 | 39.1 | 31.5 | 46.0 | 77.6 | 59.5 | 36.2 | 26.0 | 39.5 | 9.0 |
| Contriever | 44.8 | 36.0 | 49.0 | 82.3 | 56.0 | 43.7 | 31.0 | **49.5** | 13.5 |
| Dragon | 44.1 | 33.5 | 48.5 | 82.3 | 57.0 | 40.7 | 28.5 | 47.5 | 16.0 |
| OpenAI v3-large | 46.0 | 38.5 | 48.5 | 85.0 | 56.5 | 47.2 | 27.5 | 48.0 | 18.5 |
| *F1* | | | | | | | | | |
| SPScanner 1.3B | **29.2** | **20.5** | **35.1** | 48.6 | **39.9** | 26.2 | 25.6 | N/A | 12.1 |
| SPScanner 130M | **29.2** | 20.3 | 31.8 | **50.0** | 39.2 | 26.3 | 28.0 | N/A | 16.6 |
| GritLM | 28.3 | 18.3 | 33.8 | 48.7 | 39.0 | **29.0** | **29.5** | N/A | 8.8 |
| BM25 | 24.1 | 14.3 | 29.2 | 48.7 | 33.2 | 18.8 | 19.5 | N/A | 8.8 |
| Contriever | 27.0 | 17.4 | 32.8 | 49.1 | 33.0 | 24.1 | 20.0 | N/A | 11.9 |
| Dragon | 26.9 | 15.8 | 31.2 | 49.8 | 32.7 | 22.7 | 19.1 | N/A | 13.7 |
| OpenAI v3-large | 28.2 | 19.0 | 32.6 | 49.1 | 34.6 | 27.3 | 20.1 | N/A | **16.8** |
| **Llama-3.1 70B** | | | | | | | | | |
| *Accuracy* | | | | | | | | | |
| SPScanner 1.3B | **58.8** | **55.5** | 58.0 | 89.1 | **79.0** | **73.9** | 59.5 | 64.5 | 29.5 |
| SPScanner 130M | 57.5 | 52.0 | **60.0** | **90.5** | 76.5 | 67.8 | 57.5 | 60.5 | **40.0** |
| GritLM | 54.2 | 45.5 | 56.5 | 87.1 | 75.5 | 70.4 | 59.0 | **65.5** | 17.0 |
| BM25 | 46.9 | 40.0 | 50.0 | 86.4 | 72.5 | 60.8 | 48.5 | 51.0 | 5.5 |
| Contriever | 52.9 | 46.5 | 52.5 | 85.0 | 73.5 | 65.8 | 45.5 | 65.0 | 16.0 |
| Dragon | 51.9 | 44.0 | 54.0 | 85.7 | 71.0 | 60.8 | 46.0 | 58.0 | 16.0 |
| OpenAI v3-large | 55.1 | 47.5 | 54.5 | 86.4 | 74.5 | 65.3 | 50.0 | 64.0 | 22.0 |
| *F1* | | | | | | | | | |
| SPScanner 1.3B | 30.0 | **18.0** | 32.9 | **49.7** | 24.2 | **21.9** | 18.4 | N/A | 20.7 |
| SPScanner 130M | **31.8** | 16.6 | 33.2 | 49.3 | 23.6 | 20.2 | 16.9 | N/A | **27.4** |
| GritLM | 28.0 | 16.3 | 31.3 | 49.2 | 22.8 | 20.9 | 16.9 | N/A | 12.5 |
| BM25 | 23.7 | 13.1 | 29.8 | 46.8 | 23.7 | 20.0 | 15.4 | N/A | 4.0 |
| Contriever | 26.8 | 15.8 | 30.6 | 48.3 | **24.7** | 21.3 | 13.7 | N/A | 12.2 |
| Dragon | 26.6 | 15.5 | 30.7 | 48.1 | 23.0 | 19.6 | 15.1 | N/A | 12.5 |
| OpenAI v3-large | 28.2 | 17.1 | **33.4** | 47.2 | 22.9 | 19.7 | 15.6 | N/A | 16.3 |
| **Full context** | | | | | | | | | |
| *Accuracy* | | | | | | | | | |
| Llama-3.1-8B Full Context | 49.1 | 51.0 | 47.5 | 81.6 | 78.0 | 78.4 | 51.0 | 47.0 | 10.0 |
| Llama-3.1 70B Full Context | **57.8** | **56.0** | **62.0** | **85.0** | **80.5** | **85.4** | **62.5** | **65.0** | **17.0** |
| *F1* | | | | | | | | | |
| Llama-3.1-8B Full Context | 30.9 | **25.7** | **37.1** | 48.7 | **55.6** | **59.8** | **38.3** | N/A | 8.5 |
| Llama-3.1 70B Full Context | **31.3** | 20.1 | 34.0 | **48.9** | 30.4 | 39.9 | 29.2 | N/A | **12.6** |

Table 14: QA accuracy across 41 datasets with Llama-3.1 as generator with retrieval of 50 sentences. When not paired with a retriever, Llama-3.1-8B and Llama-3.1-70B are provided with the full document in-context. Results continue to next page.

| Model | longbook _qa_eng n = 200 | loogle_CR _mixup_16k n = 99 | loogle_CR _mixup_32k n = 99 | loogle_CR _mixup_64k n = 99 | loogle_CR _mixup_128k n = 99 | loogle_MIR _mixup_16k n = 139 | loogle_CR _mixup_256k n = 99 | loogle_MIR _mixup_32k n = 139 |
|---|---|---|---|---|---|---|---|---|
| | | | | **Llama-3.1 8B** | | | | |
| | | | | Accuracy | | | | |
| SPScanner 1.3B | 37.5 | 42.4 | **45.5** | **43.4** | 35.4 | **33.8** | 31.3 | 29.5 |
| SPScanner 130M | **39.0** | **44.4** | 40.4 | 36.4 | 32.3 | 30.9 | **35.4** | **34.5** |
| GritLM | 33.5 | 36.4 | 36.4 | 40.4 | 33.3 | 25.2 | 32.3 | 25.2 |
| BM25 | 19.5 | 29.3 | 35.4 | 29.3 | 31.3 | 15.1 | 25.3 | 8.6 |
| Contriever | 31.5 | 41.4 | 38.4 | 30.3 | **36.4** | 21.6 | 30.3 | 18.7 |
| Dragon | 34.5 | 38.4 | 37.4 | 29.3 | 32.3 | 22.3 | 31.3 | 21.6 |
| OpenAI v3-large | 34.5 | 36.4 | 35.4 | 38.4 | 35.4 | 20.1 | 32.3 | 25.2 |
| | | | | F1 | | | | |
| SPScanner 1.3B | 19.9 | 24.3 | 22.5 | 23.2 | **24.9** | **22.3** | 21.0 | **21.5** |
| SPScanner 130M | **20.4** | **24.6** | **23.6** | **24.5** | 23.3 | 20.9 | **22.1** | 21.1 |
| GritLM | 18.4 | 21.8 | 21.0 | 21.1 | 22.5 | 20.7 | 21.7 | 21.3 |
| BM25 | 10.8 | 19.9 | 19.7 | 20.2 | 20.7 | 13.8 | 18.4 | 13.7 |
| Contriever | 17.5 | 20.4 | 19.6 | 18.9 | 18.2 | 18.2 | 20.6 | 17.1 |
| Dragon | 19.6 | 22.4 | 19.4 | 17.6 | 17.6 | 17.1 | 21.2 | 17.8 |
| OpenAI v3-large | 18.0 | 22.4 | 20.9 | 19.8 | 21.2 | 16.8 | 20.2 | 17.9 |
| | | | | **Llama-3.1 70B** | | | | |
| | | | | Accuracy | | | | |
| SPScanner 1.3B | **56.5** | **54.5** | **50.5** | **49.5** | 44.4 | **46.8** | 38.4 | **41.0** |
| SPScanner 130M | 49.5 | 45.5 | 45.5 | 45.5 | **47.5** | 40.3 | 41.4 | **41.0** |
| GritLM | 40.5 | 44.4 | 43.4 | 40.4 | 42.4 | 34.5 | 42.4 | 36.7 |
| BM25 | 25.5 | 39.4 | 38.4 | 31.3 | 35.4 | 18.0 | 31.3 | 15.8 |
| Contriever | 40.5 | 46.5 | 44.4 | 44.4 | 40.4 | 26.6 | 41.4 | 26.6 |
| Dragon | 40.5 | 43.4 | 46.5 | 40.4 | 41.4 | 26.6 | 40.4 | 25.2 |
| OpenAI v3-large | 47.0 | 50.5 | 48.5 | 46.5 | 43.4 | 33.1 | **44.4** | 30.2 |
| | | | | F1 | | | | |
| SPScanner 1.3B | **19.5** | **22.5** | 21.0 | **22.2** | **21.4** | **22.9** | 21.6 | **22.3** |
| SPScanner 130M | 18.1 | 21.3 | 21.0 | 22.1 | 20.8 | 21.1 | 21.1 | 21.6 |
| GritLM | 16.3 | 20.1 | **21.1** | 21.8 | **21.4** | 21.0 | 18.6 | 21.3 |
| BM25 | 9.6 | 17.8 | 18.5 | 17.6 | 18.3 | 13.9 | 16.4 | 13.3 |
| Contriever | 15.2 | 20.4 | 20.0 | 19.1 | 18.2 | 16.7 | 18.2 | 16.9 |
| Dragon | 14.8 | 20.1 | 20.4 | 18.1 | 18.9 | 16.9 | 18.1 | 17.0 |
| OpenAI v3-large | 17.5 | 22.2 | 21.0 | 20.3 | 20.1 | 18.5 | **21.9** | 17.4 |
| | | | | **Full context** | | | | |
| | | | | Accuracy | | | | |
| Llama-3.1-8B Full Context | 42.0 | 42.4 | 40.4 | 27.3 | 27.3 | 31.7 | 27.3 | 32.4 |
| Llama-3.1 70B Full Context | **52.5** | **60.6** | **57.6** | **42.4** | **37.4** | **43.9** | **31.3** | **38.1** |
| | | | | F1 | | | | |
| Llama-3.1-8B Full Context | **29.2** | **25.3** | **24.8** | 22.1 | 13.5 | **26.2** | 17.6 | **23.8** |
| Llama-3.1 70B Full Context | 27.5 | 23.1 | 22.1 | **22.2** | **21.6** | 21.7 | **20.3** | 22.2 |

Table 15: Results continued

| Model | loogle_MIR _mixup_64k n = 139 | loogle_MIR _mixup_128k n = 139 | loogle_MIR _mixup_256k n = 139 | multifieldqa_en _mixup_16k n = 101 | multifieldqa_en _mixup_32k n = 101 | multifieldqa_en _mixup_64k n = 101 | multifieldqa_en _mixup_128k n = 101 |
|---|---|---|---|---|---|---|---|
| | | | | **Llama-3.1 8B** | | | |
| | | | | Accuracy | | | |
| SPScanner 1.3B | **32.4** | **30.9** | 30.9 | 51.5 | 48.5 | 51.5 | 45.5 |
| SPScanner 130M | 26.6 | 30.2 | **31.7** | 50.5 | 52.5 | 48.5 | 52.5 |
| GritLM | 24.5 | 27.3 | 23.0 | 49.5 | 47.5 | 49.5 | 44.6 |
| BM25 | 11.5 | 7.2 | 10.1 | 45.5 | 43.6 | 46.5 | 40.6 |
| Contriever | 18.7 | 21.6 | 23.0 | 48.5 | **57.4** | **57.4** | 55.4 |
| Dragon | 20.9 | 22.3 | 20.9 | **57.4** | 56.4 | 49.5 | 50.5 |
| OpenAI v3-large | 19.4 | 28.1 | 27.3 | 53.5 | 52.5 | 49.5 | **57.4** |
| | | | | F1 | | | |
| SPScanner 1.3B | **20.6** | **22.1** | 18.6 | 33.1 | 34.4 | 33.0 | 30.8 |
| SPScanner 130M | 19.7 | 19.8 | **21.2** | 31.8 | **36.1** | 32.9 | **35.5** |
| GritLM | 19.9 | 21.1 | 20.4 | 31.5 | 32.9 | 30.5 | 30.4 |
| BM25 | 13.0 | 10.2 | 11.1 | 31.9 | 29.4 | 28.5 | 29.5 |
| Contriever | 16.4 | 17.7 | 17.7 | 33.2 | 34.7 | 33.1 | 33.1 |
| Dragon | 18.1 | 16.9 | 18.1 | **34.2** | 33.9 | 33.8 | 33.9 |
| OpenAI v3-large | 18.4 | 18.7 | 18.2 | 32.5 | 35.4 | **36.9** | 34.3 |
| | | | | **Llama-3.1 70B** | | | |
| | | | | Accuracy | | | |
| SPScanner 1.3B | **41.0** | **41.7** | **39.6** | 63.4 | 63.4 | 59.4 | 62.4 |
| SPScanner 130M | 37.4 | 33.1 | 36.0 | **64.4** | 63.4 | **62.4** | **63.4** |
| GritLM | 35.3 | 35.3 | 36.0 | 57.4 | 56.4 | 54.5 | 54.5 |
| BM25 | 15.1 | 17.3 | 13.7 | 54.5 | 56.4 | 56.4 | 56.4 |
| Contriever | 26.6 | 25.9 | 22.3 | 58.4 | 61.4 | 60.4 | **63.4** |
| Dragon | 25.2 | 18.0 | 20.1 | 57.4 | **64.4** | **62.4** | 61.4 |
| OpenAI v3-large | 30.2 | 30.9 | 31.7 | 59.4 | 61.4 | 57.4 | 62.4 |
| | | | | F1 | | | |
| SPScanner 1.3B | **22.2** | **22.0** | **22.6** | 31.4 | 32.5 | 31.8 | 30.9 |
| SPScanner 130M | 20.5 | 19.7 | 20.4 | 31.1 | 31.2 | 31.9 | 31.1 |
| GritLM | 20.9 | 20.3 | 21.3 | 31.0 | 31.3 | 29.9 | 30.2 |
| BM25 | 12.6 | 12.9 | 11.0 | 28.9 | 30.1 | 28.7 | 28.3 |
| Contriever | 17.8 | 16.5 | 15.6 | 29.6 | 32.4 | 32.1 | 31.2 |
| Dragon | 17.7 | 16.3 | 16.0 | 31.2 | 31.9 | **33.0** | 31.7 |
| OpenAI v3-large | 18.8 | 17.2 | 19.1 | **32.7** | **33.4** | 32.1 | **31.8** |
| | | | | **Full context** | | | |
| | | | | Accuracy | | | |
| Llama-3.1 8B Full Context | 18.0 | 19.4 | 12.2 | 49.5 | 39.6 | 33.7 | 26.7 |
| Llama-3.1 70B Full Context | **28.8** | **23.0** | **20.9** | **58.4** | **43.6** | **39.6** | **33.7** |
| | | | | F1 | | | |
| Llama-3.1 8B Full Context | 16.8 | 13.1 | 7.1 | **37.7** | **29.1** | 25.1 | 21.8 |
| Llama-3.1 70B Full Context | **21.7** | **17.6** | **13.4** | 30.0 | 28.2 | **26.4** | **25.1** |

Table 16: Results continued

| Model | altqa _4k n = 199 | altqa _16k n = 199 | meetingpred _4k n = 100 | multifieldqa_en _mixup_256k n = 101 | meetingpred _16k n = 100 | meetingqa _4k n = 86 | meetingqa _16k n = 91 | paperqa _4k n = 82 | paperqa _16k n = 90 |
|---|---|---|---|---|---|---|---|---|---|
| | | | | **Llama-3.1 8B** | | | | | |
| | | | | Accuracy | | | | | |
| SPScanner 1.3B | **78.9** | 72.9 | **37.0** | 50.5 | **21.0** | **67.4** | 61.5 | **76.8** | 64.4 |
| SPScanner 130M | **78.9** | 72.9 | 29.0 | 45.5 | 18.0 | 65.1 | 60.4 | 73.2 | 66.7 |
| GritLM | 75.9 | **74.4** | 32.0 | 43.6 | 20.0 | 64.0 | 62.6 | 74.4 | **70.0** |
| BM25 | 62.8 | 55.3 | 27.0 | 49.5 | 18.0 | 66.3 | 64.8 | 74.4 | 68.9 |
| Contriever | 72.9 | 72.9 | 35.0 | 57.4 | **21.0** | 62.8 | 65.9 | 74.4 | **70.0** |
| Dragon | 69.8 | 66.8 | 35.0 | 56.4 | 16.0 | 60.5 | **69.2** | 69.5 | 62.2 |
| OpenAI v3-large | 75.9 | 72.4 | 36.0 | **55.4** | **21.0** | 64.0 | 64.8 | 74.4 | 65.6 |
| | | | | F1 | | | | | |
| SPScanner 1.3B | **78.5** | 71.4 | 30.8 | **33.8** | 17.8 | N/A | N/A | N/A | N/A |
| SPScanner 130M | 78.4 | 72.1 | 24.0 | 32.1 | 15.2 | N/A | N/A | N/A | N/A |
| GritLM | 75.6 | **73.4** | 26.4 | 31.3 | 16.6 | N/A | N/A | N/A | N/A |
| BM25 | 62.3 | 54.8 | 23.7 | 31.4 | 18.2 | N/A | N/A | N/A | N/A |
| Contriever | 72.4 | 72.6 | 29.5 | 33.1 | **18.8** | N/A | N/A | N/A | N/A |
| Dragon | 69.9 | 65.3 | **32.7** | 32.7 | 14.1 | N/A | N/A | N/A | N/A |
| OpenAI v3-large | 75.4 | 71.9 | 30.3 | 33.5 | 18.6 | N/A | N/A | N/A | N/A |
| | | | | **Llama-3.1 70B** | | | | | |
| | | | | Accuracy | | | | | |
| SPScanner 1.3B | 79.9 | 74.9 | **58.0** | 59.4 | **36.0** | 74.4 | 67.0 | 72.0 | **80.0** |
| SPScanner 130M | 79.9 | 73.4 | 47.0 | 60.4 | 31.0 | **76.7** | 67.0 | 74.4 | **80.0** |
| GritLM | 77.4 | 74.9 | 50.0 | 55.4 | 30.0 | 73.3 | 61.5 | 76.8 | 78.9 |
| BM25 | 68.8 | 60.3 | 42.0 | 59.4 | 16.0 | 74.4 | 64.8 | **78.0** | **80.0** |
| Contriever | 79.4 | 73.9 | 53.0 | 57.4 | 26.0 | 73.3 | **71.4** | 74.4 | **80.0** |
| Dragon | 73.9 | 70.9 | 51.0 | 58.4 | 20.0 | 75.6 | 65.9 | **78.0** | 77.8 |
| OpenAI v3-large | **81.4** | **76.4** | 45.0 | **61.4** | 34.0 | 74.4 | 67.0 | 74.4 | 75.6 |
| | | | | F1 | | | | | |
| SPScanner 1.3B | 79.0 | 74.9 | **59.2** | 31.1 | **35.4** | N/A | N/A | N/A | N/A |
| SPScanner 130M | 78.5 | 72.1 | 46.8 | 31.6 | 29.8 | N/A | N/A | N/A | N/A |
| GritLM | 76.0 | 73.9 | 48.4 | 30.6 | 28.8 | N/A | N/A | N/A | N/A |
| BM25 | 68.4 | 58.9 | 38.9 | 29.7 | 16.5 | N/A | N/A | N/A | N/A |
| Contriever | 78.5 | 73.4 | 50.8 | 30.2 | 23.9 | N/A | N/A | N/A | N/A |
| Dragon | 72.9 | 70.4 | 50.7 | 31.8 | 19.1 | N/A | N/A | N/A | N/A |
| OpenAI v3-large | **80.5** | **76.4** | 42.9 | **32.8** | 31.3 | N/A | N/A | N/A | N/A |
| | | | | **Full context** | | | | | |
| | | | | Accuracy | | | | | |
| Llama-3.1 8B Full Context | **80.4** | 77.9 | 42.0 | 32.7 | 35.0 | **82.6** | **78.0** | 81.7 | 76.7 |
| Llama-3.1 70B Full Context | 79.4 | **79.9** | **68.0** | **34.7** | **58.0** | **82.6** | 74.7 | **86.6** | **87.8** |
| | | | | F1 | | | | | |
| Llama-3.1-8B Full Context | 78.2 | 77.9 | 35.8 | 22.3 | 30.4 | N/A | N/A | N/A | N/A |
| Llama-3.1 70B Full Context | **79.4** | **79.9** | **71.3** | **23.1** | **57.2** | N/A | N/A | N/A | N/A |

Table 17: Results continued

| Model | tpo
n = 200 | financial
_qa
n = 68 | legal_contract
_qa
n = 130 | scientific
_qa
n = 161 | quality
n = 200 | coursera
n = 172 | docfinQA
n = 200 | muld
_CAC
n = 86 | ELITR
_Bench
n = 130 |
|---|---|---|---|---|---|---|---|---|---|
| | | | | **Llama-3.1 8B** | | | | | |
| | | | | Accuracy | | | | | |
| SPScanner 1.3B | 59.0 | **54.4** | **36.2** | 54.7 | 37.5 | 50.0 | **23.0** | **83.7** | 51.5 |
| SPScanner 130M | 59.5 | 45.6 | 34.6 | **57.8** | 36.0 | **52.3** | **23.0** | 80.2 | 53.8 |
| GritLM | 61.0 | 38.2 | 32.3 | 53.4 | 41.5 | 48.8 | 13.5 | 76.7 | 40.0 |
| BM25 | 60.5 | 47.1 | 23.1 | 47.8 | 37.5 | 47.7 | 0.5 | 70.9 | 56.9 |
| Contriever | **62.0** | 47.1 | 26.9 | 42.2 | **42.0** | 50.6 | 14.0 | 79.1 | 49.2 |
| Dragon | 61.5 | 50.0 | 27.7 | 46.0 | 40.5 | 43.0 | 13.5 | 82.6 | 57.7 |
| OpenAI v3-large | **62.0** | 51.5 | 35.4 | 50.9 | 39.5 | 47.7 | 22.0 | 74.4 | **59.2** |
| | | | | F1 | | | | | |
| SPScanner 1.3B | N/A | 42.6 | 23.0 | **30.4** | N/A | N/A | 3.3 | N/A | 23.7 |
| SPScanner 130M | N/A | 41.0 | **24.2** | 29.1 | N/A | N/A | **3.7** | N/A | 25.0 |
| GritLM | N/A | 41.7 | 22.8 | 30.1 | N/A | N/A | 1.5 | N/A | 23.2 |
| BM25 | N/A | 39.9 | 22.2 | 27.1 | N/A | N/A | 0.3 | N/A | **27.4** |
| Contriever | N/A | 39.0 | 20.1 | 27.2 | N/A | N/A | 2.0 | N/A | 25.2 |
| Dragon | N/A | 42.2 | 20.6 | 27.8 | N/A | N/A | 2.1 | N/A | 25.7 |
| OpenAI v3-large | N/A | **42.7** | 22.3 | 28.4 | N/A | N/A | 2.9 | N/A | 26.3 |
| | | | | **Llama-3.1 70B** | | | | | |
| | | | | Accuracy | | | | | |
| SPScanner 1.3B | 75.0 | 57.4 | 44.6 | **61.5** | 54.0 | 63.4 | **44.0** | **86.0** | 63.1 |
| SPScanner 130M | **76.0** | **58.8** | **45.4** | 60.2 | 54.0 | **65.1** | 40.5 | 84.9 | 63.1 |
| GritLM | 75.0 | 50.0 | 42.3 | 58.4 | **59.0** | 62.2 | 26.0 | 84.9 | 55.4 |
| BM25 | 74.5 | 48.5 | 27.7 | 53.4 | 53.5 | 63.4 | 3.0 | 75.6 | 58.5 |
| Contriever | 74.0 | 50.0 | 32.3 | 50.9 | 56.0 | 64.5 | 30.0 | 84.9 | 58.5 |
| Dragon | 75.0 | 51.5 | 39.2 | 54.0 | 54.5 | 61.0 | 26.5 | 84.9 | 62.3 |
| OpenAI v3-large | 74.5 | 51.5 | 42.3 | 54.0 | 54.0 | 64.0 | 35.0 | 84.9 | **66.9** |
| | | | | F1 | | | | | |
| SPScanner 1.3B | N/A | 43.5 | 24.3 | 29.6 | N/A | N/A | **2.9** | N/A | 26.2 |
| SPScanner 130M | N/A | **44.0** | **24.7** | **30.1** | N/A | N/A | 2.8 | N/A | **28.1** |
| GritLM | N/A | 42.2 | 23.6 | 29.1 | N/A | N/A | 2.0 | N/A | 24.7 |
| BM25 | N/A | 41.7 | 22.7 | 26.9 | N/A | N/A | 0.5 | N/A | 24.8 |
| Contriever | N/A | 39.4 | 21.5 | 27.0 | N/A | N/A | 2.2 | N/A | 24.8 |
| Dragon | N/A | 41.8 | 23.5 | 27.0 | N/A | N/A | 2.0 | N/A | 25.8 |
| OpenAI v3-large | N/A | 41.0 | 23.5 | 28.3 | N/A | N/A | 2.4 | N/A | 26.4 |
| | | | | **Full context** | | | | | |
| | | | | Accuracy | | | | | |
| Llama-3.1 8B Full Context | 77.0 | 67.6 | 24.6 | 56.5 | 58.0 | 65.1 | **16.5** | 84.9 | 60.8 |
| Llama-3.1 70B Full Context | **82.0** | 69.1 | **38.5** | 60.2 | 77.5 | 77.3 | 14.0 | **91.9** | **69.2** |
| | | | | F1 | | | | | |
| Llama-3.1 8B Full Context | N/A | 42.7 | 18.0 | **37.6** | N/A | N/A | **6.4** | N/A | **31.7** |
| Llama-3.1 70B Full Context | N/A | **45.0** | **25.1** | 30.9 | N/A | N/A | 1.8 | N/A | 30.3 |

Table 18: Results continued

# B Synthetic Data Generation

## B.1 Chunk-based Generation

See Figure 5 for chunk-based generation prompt.

> Given an indexed list of sentences, generate a question and answer pair from the sentences following these rules:
> 1. The question must be concrete, i.e. it should question a specific content in the given sentences.
> 2. The question and answer pair must depend on multiple sentences that spread throughout the chunk.
> 3. The answer must be coherent and concise.
>
> Indexed list of sentence: {**indexed_list_of_sentences**}
>
> First generate the question and answer. Output the question after the keyword **QUESTION:**. Output the answer after the keyword **ANSWER:**. After generating question and answer, output the indices of the sentences on which the question and answer pair depends on. Output a list of indices [index1, index2, ...] after the keyword **SENTENCES:**.

Figure 5: Prompt for chunk-based generation.

## B.2 Pair-based Generation

See Figure 6 for a pair-based generation prompt.

> Given only two pieces of information extracted from a document, invent a circumstance where there is a logical and reasonable connection between these two sentences. From this circumstance, create a question and its answer pair, such that to answer the question, one would need both pieces of information. Make sure it is IMPOSSIBLE to answer the question without knowing BOTH information. Importantly, the question must depend on this two information in a profound, non-superficial, non-apparent and NO KEYWORD OVERLAPPING way. Invent the circumstance and the connection between these two pieces of information first, output the connection after the keyword "CONNECTION:" Based on the connection, briefly explain step by step as to why it is impossible to answer the question using only one piece of information. Output all explanation and reasoning after the keyword "REASON:" Based on your reasoning and explanation, while making sure the question itself does not use ANY keyword directly from these two information, output the question after the keyword "QUESTION:" Output the answer after the keyword "ANSWER:"
>
> Information 1: {**Chunk_1**}
>
> Information 2: {**Chunk_2**}

Figure 6: Prompt for pair-based generation.

### B.3 Link-based Generation

See Figure 7 for the prompt to discover natural connections within a document.

> Given a chunk of sentences
> Chunk:
> **{chunk}**
> What other sentences in the document are highly connected to this chunk? Output each sentence index that is highly connected to the chunk, and explain the reason.
> Document:
> **{document}**

Figure 7: Prompt to discover natural connections within a document.

See Figure 8 for the synthetic question generation prompt.

> Two chunks of text in a document are connected with the following connection. Use this connection to build a question. Step by step explain how you would take advantage of this connection, and build a short, concise, one-sentence, concrete, non-conceptual, non-ambiguous question. IMPORTANTLY, you must use exact words from the connection given to you, but you must never refer to the chunks, never mention words such as "connection", "alignment", "relationship" between chunks. The question must be self-contained, and cannot not refer to the chunks, and must standalone makes sense.
>
> Output your reasoning, especially how you would take advantage of this connection, and your verification, especially why the question is non-conceptual, and concrete, and self-contained and standalone makes sense, after the keyword "REASON:"
> Based on your step-by-step reasoning and verification, then output the question after the keyword "QUESTION:"
>
> Connection:
> **{connection}**

Figure 8: Prompt for synthetic question generation based on natural connections.

See Figure 9 for labeling sentences as relevant or irrelevant prompt.

> Given a question, and a list of indexed text elements. Select the indices of all relevant text element(s) that would be helpful to answer the question.
>
> Question:
> **{question}**
>
> List of text elements:
> **{list_of_text_elements}**
>
> Recall, your task is to select indices for all relevant text elements that can help you answer the question. Provide a step-by-step explanation after the keyword 'REASON:'
> Based on your explanation, output a list of indices for text elements that are relevant and helpful to answer the question, in this format, [index1, index2, ...] ,after the keyword "LIST:"
> If no text element is helpful and relevant to answer the question, output an empty list [] after the keyword "LIST:"

Figure 9: Prompt for labeling sentences as relevant or irrelevant to a synthetic query.

### B.4 Synthetic Data Quality Evaluation

We provide and evaluate a few synthetic data examples generated from different synthetic data strategies in Tables 19, 20, and 21.

**Link-based** data (19) typically generates questions that are more coherent because these questions arise from natural connections searched within the document. The labeled sentences are implicitly linked to the question through these connections, with sentences from different parts of the document collectively forming the answer. When a model is trained on such data, identifying the first chunk can guide it to locate the second chunk. This training process teaches the model to use information from previously encountered content when evaluating the relevance of each new sentence. By training Single-Pass Scanner in this way, it learns to use its global understanding of the entire document to determine which sentences are important for answering the question.

**Human Evaluation:** The first example in Table 19 explores the significance of the little things in a marriage. In both contexts, the highlighted sentences offer intriguing insights into this topic. In the first context, a young girl asks her mother about things that might not matter in a marriage. The mother responds by emphasizing that these small details, which can take the edge off, do indeed matter. In the second context, the sentence about a husband and wife working together highlights the importance of collaboration in a marriage, i.e. *pulling together*. Thus, both contexts provide valuable information to address the question. Without the first context, the *"simple secret"* would not resonate as strongly, as it refers to the seemingly trivial yet significant details discussed by Marie and her mother. Interestingly, the female character in the second context is also the same Marie. Therefore, the first context sets the stage for the Single-Pass Scanner to identify the second context. This ability to utilize long-range connections is crucial for a deeper understanding of subsequent contexts.

The second example in Table 19 examines the tension between Lester's internal conflict with his father and his struggle to navigate societal rejection. Both contexts illuminate distinct yet interconnected aspects of this conflict. In the first context, Lester grapples with his father's disapproval and his own hesitancy to act decisively to mend their relationship. His introspection reveals his uncertainty about standing alone in the face of societal judgment. The second context depicts the external consequences of Lester's actions. Together, the two contexts demonstrate the layered nature of Lester's struggle, where his need for personal reform and decisive action is tied to both his father's approval and his standing in society. By establishing Lester's introspective conflict in the first context, Single-Pass Scanner learns to recognize and leverage this psychological groundwork when identifying relevant connections in the second context.

For chunk-based data (20), GPT-4o-mini processes each text chunk in its entirety and directly generates questions based on the information within that chunk. This approach ensures that the generated questions are highly relevant to the content, as the model can focus on the specific details present in each text segment. However, this method may lead to issues with superficial textual overlap when training the Single-Pass Scanner. The retriever might learn to search for semantically similar sentences rather than identifying deeper connections between individual sentences and the given query.

Pair-based synthetic data (21) are generated from two chunks of a long document that have high cosine similarity. High cosine similarity indicates significant textual overlap between the chunks but does not ensure logical or contextual dependencies, as demonstrated in Table 21. Consequently, the questions generated may appear unnatural. Additionally, creating questions directly from these chunks can result in questions that either consist of two merged smaller questions or are unrelated to both chunks.

The first example in 21 involves a question about an event that prompted an inquiry regarding a specific time during the group's evening activities. Context 1 effectively answers this question by discussing these activities. However, Context 2, which frequently uses keywords like "evening" and "I," creates a high semantic similarity with Context 1. Despite this similarity, Context 2 does not talk about the same event as Context 1 and does not contribute to answering the question in any sense. Similarly, the second example inquires about Thomas's motivation for confessing. Context 1 clearly explains that Thomas confessed

because he felt sorry for Mary. In contrast, Context 2 is unrelated to the question; it only contains negative words that might have some semantic similarity to the question.

| | **Question/Context; Important Sentences Highlighted** | **Connection** |
|---|---|---|
| Example 1 | *Question:* What are some little things that matter in a marriage?

*Context 1:* "I shan't own anything of the kind till you've been married three months, and he's had some bad dinners, and late breakfasts, and has got a bit sick of the butcher's bill. Then we'll see.""Little things like these can't matter between people who really love each other. You don't understand." "It's just these little things that take the edge off. "Marie's mother looked in and smiled to see her girl fingering her pretty things.

*Context 2:* they had made their beds and made them wrong; the great thing, the simple secret, was to make them right.A husband and wife must pull together, in everything. Pulling together would be sheer joy."Osborn," she said, "how well we understand each other, don't we?""I should think we do," whispered the young man. "Few married people seem really happy.""They must manage life badly, mustn't they?" "I remember mother and father; mother likes the idea of my getting married, but they used often to be nagging about something. | This sentence highlights the connection and understanding between partners in a marriage, which resonates with the chunk's exploration of love and the little things that matter in a relationship. |
| Example 2 | *Question:* How does Lester's internal conflict regarding his relationship with his father influence his need for decisive action in the face of social rejection and the need for reform?

*Context 1:* It was a long time before he stirred.And still, in the bottom of his heart, his erring son continued to appeal to him.CHAPTER XL Lester returned to Chicago. He realized that he had offended his father seriously, how seriously he could not say.In all his personal relations with old Archibald he had never seen him so worked up. But even now Lester did not feel that the breach was irreparable; he hardly realized that it was necessary for him to act decisively if he hoped to retain his father's affection and confidence. As for the world at large, what did it matter how much people talked or what they said. He was big enough to stand alone.But was he?People turn so quickly from weakness or the shadow of it.

*Context 2:* or at least the more conservative part of it would not.There were a few bachelors, a few gay married men, some sophisticated women, single and married, who saw through it all and liked him just the same, but they did not make society.He was virtually an outcast, and nothing could save him but to reform his ways; in other words, he must give up Jennie once and for all.But he did not want to do this. The thought was painful to him–objectionable in every way.Jennie was growing in mental acumen. She was beginning to see things quite as clearly as he did.She was not a cheap, | This sentence highlights Lester's internal conflict regarding his relationship with his father and the need for decisive action, which connects to the chunk's theme of social rejection and the need for reform. |

Table 19: Linked-based Synthetic Data Examples

| | **Question/Context; Important Sentences Highlighted** |
|---|---|
| Example 1 | *Question:* What happens to previously granted Incentive Awards after the termination of the Plan?

*Context:*
6.1 EFFECTIVE DATE AND GRANT PERIOD

This Plan shall be effective as of the date of Board approval, March 24, 1998. Unless sooner terminated by the Board, the Plan shall terminate on March 24, 2008, unless extended. After the termination of the Plan, no Incentive Awards may be granted under the Plan, but previously granted awards shall remain outstanding in accordance with their applicable terms and conditions. |
| Example 2 | *Question:* What does Mr. Pennimore emphasize about the purpose of Gerald's time at the school?

*Context:* Dan nodded."You'd better believe he does! If he says you can't play baseball or football you can't, and that's all there is to it. But he's square, all right, is 'Muscles,' and you want to do just as he tells you. He's a wonder!" Gerald considered this in silence a moment. Then: "If a fellow can't play baseball and things I don't see any use of coming here," he murmured. Mr.Pennimore laughed. "So that's your idea, is it, son? Well, let me tell you that you're here to fit yourself for college. You wanted to come here, Gerald, and you've had your way. Now there must be no backing down, my boy. Life isn't all play, as you'll find out when you get older, but you can make it seem like play by taking an interest in work.You mustn't think that because I've got money enough for us both that you're going to sit down and twiddle your thumbs and watch the procession go by. No, sir!You're going to march with the rest, and I want to see you marching at the head. |

Table 20: Chunk-based Synthetic Data Examples

| | Question/Context; Important Sentences Highlighted |
|---|---|
| Example 1 | *Question:* What significant event occurred that prompted a query about a specific time during the group's evening activities?

*Context 1:*
I walked to the house of a banker who entertained me.Naturally, my evening thoughts reverted to my home, and after reading a few verses in my Testament, I walked about the room until nearly eleven, thinking of my wife, and breathing the prayer, 'God bless you.' "I might not have recalled all the circumstances, save for the letter I received by the next post from her, with the query put in: 'Tell me what you were doing within a few minutes of eleven o'clock on Friday evening?I will tell you in my next why I ask; for something happened to me.'In the middle of the week the letter came, and these words in it:–'I had just awoke from a slight repose, when I saw you in your night-dress bend over me, and utter the words, "God bless you!"I seemed also to feel your breath as you kissed me.'

*Context 2:*
I was deputed along with a medical officer to proceed to the nearest railway station at that time Allahabad, in charge of a sick officer.I will call myself Brown, the medical officer Jones, and the sick officer Robertson.We had to travel very slowly, Robertson being carried by coolies in a doolie, and on this account we had to halt at a rest-house, or pitch our camp every evening.One evening, when three marches out of Banda, I had just come into Robertson's room about midnight to relieve Jones, for Robertson was so ill that we took it by turns to watch him, when Jones took me aside and whispered that he was afraid our friend was dying, that he did not expect him to live through the night, and though I urged him to go and lie down, and that I would call him on any change taking place, he would not leave.We both sat down and watched. |
| Example 2 | *Question:* What motivated Thomas to seek forgiveness and confess his past actions?

*Context 1:*
"Oh, no!He's a gen—" but was drowned in laughter.He threw his head up and laughed to the sky."You're a wonder, I must say.I beg him ten thousand pardons—I forgot.Of course, he's a gentleman."
Mary was piqued."That's not very kind of you," she said, with reproach in her tones, and he humbled himself at once."I'm very sorry, but I'll confess the whole.The fact is, you've jumped into a little pit which I had dug for you—headlong.Upon my word, I beg your pardon.But don't you know that these class-boxes into which you plump every mother's son of us, and are at such pains to keep guarded, lest one of us should step out, are the very things I'm vowed to destroy?

*Context 2:*
Only, when desire fades in us, o' God's name let us die.Our friend here cried in his heart that his had never bloomed before.Spell-bound to a beautiful vision, he walked enraptured in the light of it, travelling up the path of its beam, sighing, not that it should be so long, but that his steps should lag so short of his urgency.And to the lips of his heart—as it were—recurred and recurred the dear, familiar phrases, true once and true now to who so love.The well-found hearth, and One beside it: surely, happily there!Denied him for so long; now in full sight!The buffeting, windy world outside, the good door barred, the ruddy fire, the welcoming arms, the low glad voice! |

Table 21: Pair-based Synthetic Data Examples

## C    Test Set Evaluation

### C.1    Freeform Question-Answer Judging Prompt

See Figure 10 for an freeform question-answer judging prompt.

---

Given a question, a groundtruth answer, and an attempted answer, use the following criteria to determine if the attempted answer accurately reflects the groundtruth answer.
Criteria:
- The majority of the information in the attempted answer should overlap with the groundtruth answer. Note that the attempted answer may include additional information derived from the question.
- The attempted answer may extend the groundtruth answer while covering all its aspects, however, the attempted answer should not be contradicting the groundtruth answer.
- If the groundtruth contains numbers, the attempted answer must match when rounded to the same precision as the groundtruth.
Example 1:
Groundtruth Answer: 1983
Attempted Answer: 1.983 million
Reason: The groundtruth answer 1983 is a whole number without any units. The attempted answer uses 1.983, which is different from the whole number 1983 and thus should be considered incorrect.
Decision: NO
Example 2:
Groundtruth Answer: 93
Attempted Answer: 93 million
Reason: The attempted answer is 93 million, which uses the same digits as the attempted answer and thus should be considered correct.
Decision: YES
Question:
**{question}**
Groundtruth Answer:
**{gt_answer}**
Attempted Answer:
**{answer}**
Think step by step when you compare these two answers. Based on the reasoning, output a YES/NO decision after the keyword "DECISION:".

---

Figure 10: Prompt for judging the correctness of an answer to an freeform question.

### C.2    Multiple Choice Question Question-Answer Judging Prompt

See Figure 11 for multiple choice question-answer judging prompt.

---

Given a multiple-choice question, a ground truth answer, and an attempted answer, the attempted answer should be the same option as the ground truth answer. It should not include any other options beyond those in the ground truth answers. Some parts of the attempted answer may overlap with information from the question.

Question:
**{question}**
Ground Truth Answer:
**{gt_answer}**
Attempted Answer:
**{answer}**
Think step by step when you compare these two answers. Based on the reasoning, output a YES/NO decision after the keyword "DECISION:".

---

Figure 11: Prompt for judging the correctness of an answer to a multiple choice question.

## C.3 Relevant Sentence Annotation Prompt

See Figure 12 for the relevant sentence annotation prompt.

Given a question and its groundtruth answer, and a list of indexed text elements. Select the indices of all relevant text element(s) that would help derive or at least infer the groundtruth answer to the question from the context.

Question:
**{question}**

Groundtruth answer:
**{gt_answer}**

list of text elements:
**{list_of_text_elements}**

Recall your task is to determine if you can derive or at least infer the groundtruth answer to the question from the list of text elements, and then select indices for all relevant text elements.
Provide an explanation after the keyword 'REASON:'
Based on your explanation, output a YES/NO decision after the keyword 'DECISION:'
Lastly, if your decision is YES, output a list of indices for text elements that are needed to derive the groundtruth answer in this format, [index1, index2, ...] ,after the keyword "LIST:"

Figure 12: Prompt for annotating relevant sentences for the given query.

## D Training Hyperparameter Setting

Our training process used a peak learning rate of $1 \times 10^{-4}$, optimized on the validation set, and a minimum learning rate of $1 \times 10^{-5}$. We used an effective batch size of 64 by setting the gradient accumulation steps to 8 and applied a maximum gradient norm of 1. Optimization was performed using the AdamW optimizer (Loshchilov & Hutter, 2019) with $\beta = (0.9, 0.95)$ and a weight decay of 0.01. A cosine learning rate scheduler with a 10% warmup phase was employed. Additionally, mixed-precision training with BF16 was utilized to enhance computational efficiency and reduce exploding gradient issue.

# E   Further Analyses

## E.1   Model Performance on Scientific Documents

| Retrievers with GPT-4o as Generator | Benchmark Datasets | | | | |
|---|---|---|---|---|---|
| | Qasper $n = 200$ | Scientific_QA $n = 161$ | Paper_QA (4k) $n = 82$ | Paper_QA (16k) $n = 90$ | Total $n = 533$ |
| BM25 | 38.5 | 41.6 | 81.7 | 82.2 | 53.5 |
| Contriever-110M | 48.5 | 50.9 | 80.5 | 84.4 | 60.2 |
| Contriever-110M-FT | 49.5 | 54.0 | 84.1 | 85.6 | 62.3 |
| GTE-Qwen2-1.5B | 49.5 | 57.1 | 81.7 | 85.6 | 62.8 |
| Stella-1.5B | 51.0 | 50.7 | 86.6 | 82.2 | 61.7 |
| GritLM-7B | 49.5 | 54.0 | 86.6 | 86.7 | 62.8 |
| NV-Embed-v2-7B | 52.5 | 54.7 | 79.3 | 86.7 | 63.1 |
| SPScanner 130M | 53.5 | 59.0 | 80.5 | 85.6 | 64.7 |
| SPScanner 1.3B | 57.5 | 59.6 | 85.4 | 85.6 | 67.2 |
| GPT-4o Full Context | 58.5 | 62.7 | 84.1 | 83.3 | 67.9 |

Table 22: Models' performance on four benchmarks containing scientific document.

## E.2   Discriminative Models vs. Generative Models

| Model Type | Model | Average Accuracy |
|---|---|---|
| Generative | GPT-4o | 52.2 |
| | Llama-3.1 70B | 45.9 |
| | Mamba-2 130M-FT | 33.5 |
| Discriminative | SPScanner 130M | 60.0 |
| | SPScanner 1.3B | 61.8 |

Table 23: Average Accuracy is based on all data points from 41 test sets. FT means fine-tuned. Note, Mamba-2 130M-FT is a generative model, whereas SPScanner 130M is our proposed discriminative model.

Single-Pass Scanners are discriminative models that use a classification head to score each sentence. We investigate the feasibility of using an LLM to generatively select sentences via next token prediction. We evaluate models on all test sets and report average accuracy. Given a full document, GPT-4o and Llama-3.1-70B are instructed to select relevant sentences for up to 2000 tokens, which is a fair comparison with the "50 sentences" setup in Single-Pass Scanners. Due to poor generative performance of the pre-trained Mamba-2 checkpoint, we fine-tune Mamba-2-130M to generate relevant sentences using the same 1 million link-based synthetic data. Table 23 shows that all generative models are significantly worse than discriminative models. This suggests the generative approach is often lossy in long-context setting.

## E.3   Direct Answer Generation on Full Context

| Model | GPT-4o | Llama-3.1 | | Mamba-2 | | | |
|---|---|---|---|---|---|---|---|
| | | 70B | 8B | 130M-FT | 130M | 1.3B-FT | 1.3B |
| Average Accuracy | 64.6 | 57.8 | 49.1 | 15.6 | 0.56 | 27.6 | 0.59 |

Table 24: Direct answer generation from full context.

In Table 24, we investigate whether LLMs such as GPT-4o, Llama-3.1-70B, Llama-3.1-8B, Mamba-2-1.3B, and Mamba-2-130M are able to directly answer questions based on the full context of long documents.

GPT-4o has the highest accuracy on test sets. Llama-3.1-70B and Llama-3.1-8B achieve worse performance than SPScanner 1.3B or 130M. Note, Table 2 shows Llama-3.1-70B uses 28.5 PFLOPs while SPScanner 130M uses 19 TFLOPs.

Pretrained Mamba-2 models are unable to answer questions based on long documents. We fine-tuned the Mamba-2-1.3B and 130M models on the same 1 million link-based data with answers generated by GPT-4o-mini and a conditional language modeling objective. While these fine-tuned checkpoints are considerably better than pretrained counterparts, they have substantially lower performance than Single-Pass Scanners. This shows it is challenging to train state-space models directly for answer generation on long documents.

### E.4 Ablation for Relative Position of Linked-Chunks

We investigate whether the relative positions of chunks (i.e., labeled sentences) impact the training and performance of Mamba models. From the 1 million link-based synthetic data points, we select instances where the relative positions of both chunks (with respect to the full document) fall within the first 33%, between 33% and 67%, and after 67%. For each group, we randomly select 100k data points to train the SPScanner 130M model. From Table 25, we observe an incremental pattern in Single-Pass Scanner's performance: it is worst when the chunks are located in the first third of the document, improves when the chunks are situated between the first and second thirds, and is best when the chunks are positioned after the second third. This pattern may be due to the increasing distance from the query (at the beginning of the document); the further apart the labeled sentences are from the query, the more challenging the training data becomes, leading to better performance for Mamba.

| Linked-Chunks' Relative Positions | Document Type | | | | |
| --- | --- | --- | --- | --- | --- |
| | Educational *n=1967* | Creative *n=1733* | Official *n=1328* | Conversational *n=707* | Average Accuracy |
| Both in 0-33% of the document | 55.8 | 27.5 | 38.3 | 37.5 | 39.5 |
| Both in 33-67% of the document | 56.5 | 30.6 | 41.8 | 38.9 | 42.0 |
| Both in 67-100% of the document | 63.0 | 41.5 | 49.8 | 45.1 | 50.9 |

Table 25: Ablation study for the relative positions of the two linked chunks in a document using Mamba-2-130M trained on 100k data.

### E.5 Ablation for Training Document Length

| Mamba-2-130M trained on 600m tokens | | | Document Type | | | | |
| --- | --- | --- | --- | --- | --- | --- | --- |
| Input Sequence Length | | | Educational | Creative | Official | Conversational | Average |
| 2k tokens | 5k tokens | 10k tokens | *n = 1967* | *n = 1733* | *n = 1328* | *n = 707* | Accuracy |
| 300k data | - | - | 62.2 | 34.9 | 50.0 | 41.3 | 47.2 |
| 86k data | 86k data | - | 64.9 | 41.4 | 52.6 | 43.4 | 51.6 |
| 35k data | 35k data | 35k data | 66.9 | 49.3 | 57.8 | 47.1 | 56.4 |

Table 26: Ablation study for the training document sequence length.

We study whether the training document length has an impact on the performance of Single-Pass Scanners. We designed three training sets, each with a total of 600 million tokens. The first set purely contains documents of 2k tokens. The second set contains half 2k-token documents and half 5k-token documents. The third set contains an equal amount

of 2k-token, 5k-token, and 10k-token documents. From Table 26, we see the training set where 2k, 5k and 10k-token documents are mixed leads to the best Mamba performance. However, when we increase document length to 15k tokens, we observe unstable gradient norm behaviors that lead to quickly deteriorating performance of Mamba on validation sets, similar to the exploding gradient issue reported in state-space models Gu & Dao (2024); Dao & Gu (2024).

### E.6 Are Single-Pass Scanners Lost in the Middle?

"Lost in the Middle" is a phenomenon identified by Liu et al. (2024a), where large language models (LLMs) tend to lose track of information in the middle of a long document, favoring information at the beginning and end. To investigate the behavior of Single-Pass Scanners when processing long documents, we first identify useful and important information within a document and record their positions. We then examine whether Single-Pass Scanners are more likely to forget or ignore important information from specific locations within long documents.

We designed an LLM-powered pipeline that scans through a long document using a sliding window of 200 sentences, with a stride of 100 sentences. The goal is to identify all sentences potentially relevant to providing the ground-truth answer to a given question. We use GPT-4o, supplying it with both the question and the reference ground-truth answer for all data points with document lengths up to 120k tokens. Since the main paper employs a sliding window approach to aggregate logits produced by Single-Pass Scanners, it is not practical to investigate potential "lost in the middle" issues for documents exceeding 120k tokens.

With knowledge of both the question and the ground-truth answer, GPT-4o is better equipped to identify relevant sentences within a 200-sentence window. The sliding window approach is designed to mitigate potential long-context issues with GPT-4o.

After GPT-4o identifies relevant sentences in each window, we aggregate these sentences from different windows and present them to GPT-4o for a final selection. Once GPT-4o selects a final list of sentences, we ask it again whether these sentences can yield the correct ground-truth answer to the question. This step serves as a filtering process. After filtering, we have 3,067 data points with documents under 120k tokens. We manually reviewed a random subset of 200 data points to validate the quality of this pipeline.

We now have a set of 3,067 data points with documents annotated for relevant sentences. We also have SPScanner 1.3B top 50 retrieved sentences for each of these data points. For each relative position, we calculate the number of relevant sentences retrieved by SPScanner 1.3B, divided by the total number of relevant sentences found in that position. This metric is known as sentence recall at a certain relative position. Note that relative position is used because documents vary in length.

In Figure 13, we observed an interesting pattern. Single-Pass Scanner's recall performance is noticeably better for smaller relative positions (i.e., the beginning of the document). Single-Pass Scanner's recall performance drops to its lowest for the last 10% of relative positions (i.e., the end of the document). We also observed a general decreasing trend in Single-Pass Scanner's recall as the relative position increases. This suggests that the Single-Pass Scanner is less effective at retrieving relevant sentences when they are located farther from the beginning of the document (i.e., where the query is). While there is no discernible "lost in the middle" pattern in Figure 13, we did find that Single-Pass Scanner tends to lose track at the end of the document.

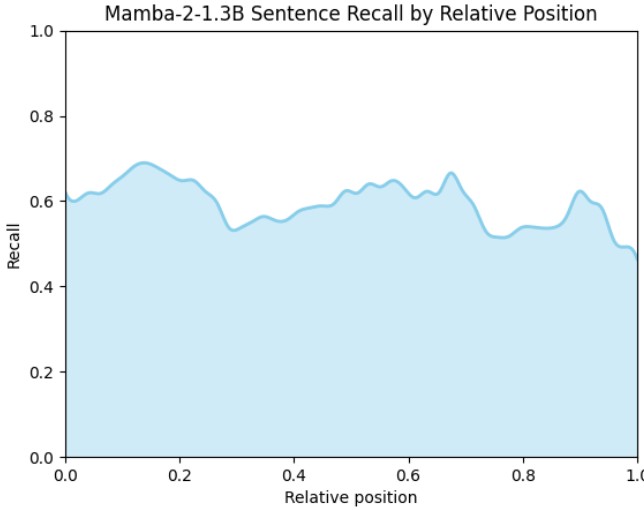

Figure 13: The recall of SPScanner 1.3B at different relative positions.

### E.7 Retrieval Comparison between Single-Pass Scanner and NV-Embed-v2

We demonstrate that the Single-Pass Scanner retrieves more relevant context than the embedding model by comparing the sentences retrieved by SPScanner 1.3B and NV-Embed-v2-7B for the following data points in the test set (examples where NV-Embed-v2-7B retrieves 50 sentences are in Tables 28, 29; examples where NV-Embed-v2-7B retrieves 5 chunks are in Tables 30).

In Table 28, NV-Embed-v2-7B successfully retrieves the semantically relevant sentence *"Uh, like a test of availability,"* which aligns with the query about *"test components' availability."* However, NV-Embed-v2-7B failed to retrieve a crucial follow-up sentence identifying PERSON 7 as the individual responsible for the test. This limitation highlights that, while NV-Embed-v2-7B effectively identifies phrases with high semantic similarity, it failed to capture broader contextual relationships needed for comprehensive information retrieval. In contrast, the SPScanner 1.3B demonstrated a stronger contextual understanding by successfully retrieving the sentence *"So that's another thing that, that [PERSON7], uh, uh, uh, should <unintelligible> on,"* which was crucial for fully answering the question.

The example in Table 29 reveals NV-Embed's ability to handle conversational text. The model retrieves relevant dialogue between Castiel and Mr.Soren, where the conversation includes several keyword matches, such as "book," "Castiel," and "Mr. Soren" found in the query. However, identifying the book as "The History of the Devil" requires a deeper contextual understanding, as this connection established in earlier parts of the conversation. The SPScanner 1.3B demonstrates this capability by successfully retrieving the key sentence: *"The 'History of the Devil,' by Daniel Defoe,-not quite the right book for a little girl."* Additionally, with prior context, the SPScanner 1.3B also retrieves *"I advise you to put by the 'History of the Devil'."* This additional context enables the generator model to provide a more accurate response to the query about the conversation.

However, overall outperforms SP-Scanner on conversational datasets. NV-embed's superior performance on conversational datasets can be attributed to the specific nature of these tasks, which often do not require capturing long-range dependencies from distant parts of the document. Since embedding models can retrieve local context but falter at long-range dependencies, these conversational datasets avoid the weakness of embedding models. Of the 707 data points in the conversational datasets, 400 focus specifically on speaker identification tasks, divided into two categories: 200 questions about masked name entities and 200 requiring identification of speakers for given sentences. Both tasks only require

knowing the immediate context to answer the question. We provide one example for each task in Table 27 below.

| **Example 1: Masked Name Entity** | |
|---|---|
| **Task** | Masked Name Entities |
| **Question** | Which character is $$MASK$$? |
| **Context** | PATRICK: I do not care for your language. LEONARD: I'm only trying to help you guys. Leonard stops exercising. $$MASK$$: (CONT'D) You're fat dad. Mom is fat. Us kids are fat. (CONTINUED). CONTINUED: PATRICK: What's your point? |
| **Answer** | LEONARD |
| **Example 2: Speaker Identification** | |
| **Task** | Speaker Identification |
| **Question** | 'Who would like to come in on that? Go on, Catherine.' said by ___ |
| **Context** | "And Catherine, I just wonder, extending Alun's point, . . ." said by Huw Irranca-Davies AM "On the point about being boring, generally boring is good as far as the EU is concerned . . ." said by Alun Davies AM "Who would like to come in on that? Go on, Catherine." said by ___ |
| **Answer** | Huw Irranca-Davies AM |

Table 27: Examples of retrieval tasks

Table 30 compares the retrieval performance of SPScanner 1.3B and NV-Embed. In this comparison, Single-Pass Scanner retrieves 50 sentences from the document, while NV-Embed-v2-7B retrieves 5 chunks. While NV-Embed's retrieved chunks contain multiple mentions of "MAVERICK" and "ROOSTER" that are semantically relevant to the query, the model misses crucial sentences that describe Rooster's frustration with Maverick for withdrawing his Naval Academy application. Specifically, in the "5 chunks" setting, NV-Embed-v2-7B fails to retrieve a key section where Rooster explicitly states: *"Maverick. He pulled my papers. He pulled my application to the Naval Academy. He set me back four years."* This omission results in the generator model producing a less accurate and incomplete response, as these sentences directly answer the query and provide vital context about the strained relationship between Rooster and Maverick. In contrast, SPScanner 1.3B successfully captures both the sentences that explicitly describe Rooster's frustration and Maverick's actions. As a result, the generator gives an attempted answer that aligns more closely with the reference answer.

Table 31 demonstrates another comparison between the Single-Pass Scanner and NV-Embed-v2-7B when retrieving the top 5 chunks. The question asks about the age of Aelis' first husband and Birdy's father. To answer the question accurately, the model needs to retrieve sentences indicating that "LORD ROLLO" is Birdy's father. As shown in the left column, the Single-Pass Scanner successfully retrieves this information and, using this context, retrieves the relevant sentence stating "Lord Rollo - 41 years of age." In contrast, the right column shows that NV-Embed-v2-7B retrieves a highly relevant-seeming chunk containing phrases such as "LORD SIDEBOTTOM, Aelis' father," "LORD GIDEON SIDEBOTTOM - 81 years of age - oldest man in his province - oldest father," and "Birdy." While this chunk includes information about "Aelis," "Birdy," ages, and "father," it fails to answer the specific question at hand. In the final portion of the text, both retrievers successfully obtain information about Aelis' husband's age. However, the key difference is that the LLM can provide a correct answer using the Single-Pass Scanner's results, while it can only make an educated guess based on the incomplete information retrieved by NV-Embed.

| | |
|---|---|
| *Question:* Who is in charge of writing the code to test components' availability during live demos? *Reference Answer:* [PERSON7]. | |
| **SPScanner 1.3B** | **NV-Embed** |
| *Attempt:* [PERSON7] is in charge of writing the code to test components' availability during live demos. | *Attempt:* [PERSON13] is in charge of writing the code to test components' availability during live demos. |
| That we would know, uh, which of the parts of the pipeline are, performing badly in terms of translation quality.Uh, I just, uh-.It just occurred to me that there should be one more compilation target. And that would be like probing whether the components of the pipeline are up and running.Uh, like a test of availability. So that's another thing that, that [PERSON7], uh, uh, uh, should <unintelligible> on-. if you could put this on, uh, on the to do list or on the enhancement options, that would also be very useful.Uh, and another thing would be, uh, like live debugging, uh, of a of a pipeline such a speed of, uh, of that.Okay.And, uh, yes, and uh, so, and then the second item you have in your list [PERSON7].Please comment on that.(PERSON7) All, right.So, uh, next Friday, uh, like, next week somewhere there there is going to be a conference about [PROJECT13] and we are going to provide life subtitles and transcription.And because we will have some non native English speakers in there, so we will need to get some feedback, from the people that are using our subtitles.Preferab-, refe-, preferably life.So we can, uh, see a moments like, uh, where it was working and moments where it was not working.So, uh ,I will make some, uh, quick took-.(PERSON13) [PERSON15] already has such simple tool that you could adapt.Um, what is more-.What is missing is, uh, description, of, ah, like how to use the tool.And also more like a generic description of how people should look at the outputs.So, uh, it would be best, if you could get in touch with [PERSON18].Because I've asked [PERSON18] to like handle the soft, uh, soft things with the participants and also with the organisers.And, uh, um, you and [PERSON18] should prepare very simple instructions that the participants could follow. | That we would know, uh, which of the parts of the pipeline are, performing badly in terms of translation quality.Uh, I just, uh-.It just occurred to me that there should be one more compilation target. And that would be like probing whether the components of the pipeline are up and running.Uh, like a test of availability. So that's another thing that, that [PERSON7], uh, uh, uh, should <unintelligible> on-.. if you could put this on, uh, on the to do list or on the enhancement options, that would also be very useful.Uh, and another thing would be, uh, like live debugging, uh, of a of a pipeline such a speed of, uh, of that.Okay.And, uh, yes, and uh, so, and then the second item you have in your list [PERSON7].Please comment on that.(PERSON7) All, right.So, uh, next Friday, uh, like, next week somewhere there there is going to be a conference about [PROJECT13] and we are going to provide life subtitles and transcription.And because we will have some non native English speakers in there, so we will need to get some feedback, from the people that are using our subtitles.Preferab-, refe-, preferably life.So we can, uh, see a moments like, uh, where it was working and moments where it was not working.So, uh ,I will make some, uh, quick took-.(PERSON13) [PERSON15] already has such simple tool that you could adapt.Um, what is more-.What is missing is, uh, description, of, ah, like how to use the tool.And also more like a generic description of how people should look at the outputs.So, uh, it would be best, if you could get in touch with [PERSON18].Because I've asked [PERSON18] to like handle the soft, uh, soft things with the participants and also with the organisers.And, uh, um, you and [PERSON18] should prepare very simple instructions that the participants could follow. |

Table 28: Example 1: Comparison of retrieval results between the SPScanner 1.3B and NV-Embed-v2-7B in the "50 sents" setting. A portion of the document is displayed, with retrieved sentences highlighted for both models in yellow. Information important for answering the question but missed by NV-Embed-v2-7B is highlighted in red in the text.

| *Question:* What book does Castiel show Mr. Soren that she is reading? | |
|---|---|
| *Reference Answer:* "The History of the Devil". | |
| **SPScanner 1.3B** | **NV-Embed** |
| *Attempt:* The 'History of the Devil,' by Daniel Defoe. | *Attempt:* "History of the Decline and Fall of the Roman Empire." |
| Mr. Roberta had listened to this exposition of Castiel's with petrifying wonder. "Why, what book is it the wench has got hold on?"he burst out at last. The 'History of the Devil,' by Daniel Defoe,–not quite the right book for a little girl, " said Mr. Soren."How came it among your books, Mr.Roberta?" Castiel looked hurt and discouraged, while her father said,– "Why, it's one o' the books I bought at Partridge's sale. They was all bound alike,–it's a good binding, you see,–and I thought they'd be all good books. There's Sara Taylor's 'Holy Living and Dying' among 'em.I read in it often of a Sunday" (Mr. Roberta felt somehow a familiarity with that great writer, because his name was Sara); "and there's a lot more of 'em,–sermons mostly, I think,–but they've all got the same covers, and I thought they were all o' one sample, as you may say.But it seems one mustn't judge by th' outside.This is a puzzlin' world." "Well," said Mr. Soren, in an admonitory, patronizing tone as he patted Castiel on the head, "I advise you to put by the 'History of the Devil,' and read some prettier book. Have you no prettier books?""Oh, yes," said Castiel, reviving a little in the desire to vindicate the variety of her reading. "I know the reading in this book isn't pretty; but I like the pictures, and I make stories to the pictures out of my own head, you know.But I've got 'AEsop's Fables,' and a book about Kangaroos and things, and the 'Pilgrim's Progress.'""Ah, a beautiful book," said Mr. Soren; "you can't read a better." | Mr. Roberta had listened to this exposition of Castiel's with petrifying wonder. "Why, what book is it the wench has got hold on?"he burst out at last. "The 'History of the Devil,' by Daniel Defoe,–not quite the right book for a little girl, " said Mr. Soren."How came it among your books, Mr.Roberta?" Castiel looked hurt and discouraged, while her father said,– "Why, it's one o' the books I bought at Partridge's sale. They was all bound alike,–it's a good binding, you see,–and I thought they'd be all good books. There's Sara Taylor's 'Holy Living and Dying' among 'em.I read in it often of a Sunday" (Mr. Roberta felt somehow a familiarity with that great writer, because his name was Sara); "and there's a lot more of 'em,–sermons mostly, I think,–but they've all got the same covers, and I thought they were all o' one sample, as you may say.But it seems one mustn't judge by th' outside.This is a puzzlin' world." "Well," said Mr. Soren, in an admonitory, patronizing tone as he patted Castiel on the head, "I advise you to put by the 'History of the Devil,' and read some prettier book. Have you no prettier books?""Oh, yes," said Castiel, reviving a little in the desire to vindicate the variety of her reading. "I know the reading in this book isn't pretty; but I like the pictures, and I make stories to the pictures out of my own head, you know.But I've got 'AEsop's Fables,' and a book about Kangaroos and things, and the 'Pilgrim's Progress.'""Ah, a beautiful book," said Mr. Soren; "you can't read a better." |

Table 29: Example 2: comparison of retrieval results between the SPScanner 1.3B and NV-Embed-v2-7B in the "50 sents" setting. A portion of the document is displayed, with retrieved sentences highlighted for both models in yellow. Information important for answering the question but missed by NV-Embed-v2-7B is highlighted in red in the text.

| | |
|---|---|
| *Question:* Why do Rooster hate MAVERICK? | |
| *Reference Answer:* Because MAVERICK pulled Rooster's application to the Naval academy. | |
| **SPScanner 1.3B** | **NV-Embed** |
| *Attempt:* Rooster hates Maverick because Maverick pulled his application to the Naval Academy, setting him back four years, which Rooster sees as an unjust hindrance to his career. | *Attempt:* Rooster hates Maverick because he blames him for the incident involving his father's death. |
| SKIES - SORTIE 4 114 114 MAVERICK (TO SELF) Sorry, Rooster.[MAVERICK LEVELS OUT, STRIKES WITH A COBRA MANEUVER, FORCING ROOSTER AND HANGMAN TO SPLIT AND OVERSHOOT HIM.] Now Mav s instantly in chase position for a shot of his own.INT. ROOSTER'S F-18 - SORTIE 4 115 115 Rooster hears the tone. MAVERICK That s a kill.INT.ROOSTER S F-18 - SORTIE 4 116 116 Rooster seethes, outwitted, but concedes the fight... ROOSTER Copy kill.INT.READY ROOM - SORTIE 4 117 117 Everyone exhales, shares a collective look.This is next level shit, even for them.EXT.TARMAC - ELSEWHERE - DUSK 118 118 Close on Rooster, sweating and furious as he does push-ups on the tarmac, punishing himself. (CONTINUED)CHERRY 11.25.19 - OFFICIAL 62. 8FLiX.com FYC SCREENPLAY DATABASE 20221226HONDO Alright. That s enough man.Rooster, that s enough.Hondo pats Rooster on the shoulder. HONDO (CONT D) Tomorrow s another day.Rooster sits up, exhausted.Feet appear next to him.He looks up to see Phoenix above him.PHOENIX What is going on with you? You trying to get kicked out?Breaking the hard deck.Insubordination.That wasn t you up there. Talk to me.What s up?ROOSTER Don t worry about it. PHOENIX I m going on this mission.But if you get kicked out, you could leave us flying with Hangman.So what the hell was that- ROOSTER HE PULLED MY PAPERS. PHOENIX What?Who?ROOSTER Maverick. He pulled my application to the Naval academy. He set me back four years.Phoenix processes.PHOENIX Why would he do that? Rooster does not answer.INT.READY ROOM 119 119 Hangman is staring at something on the wall. HANGMAN Yo, Coyote.CONTINUED: 118 118 (CONTINUED)CHERRY 11.25.19 - OFFICIAL | SKIES - SORTIE 4 114 114 MAVERICK (TO SELF) Sorry, Rooster.[MAVERICK LEVELS OUT, STRIKES WITH A COBRA MANEUVER, FORCING ROOSTER AND HANGMAN TO SPLIT AND OVERSHOOT HIM.] Now Mav s instantly in chase position for a shot of his own.INT. ROOSTER'S F-18 - SORTIE 4 115 115 Rooster hears the tone. MAVERICK That s a kill.INT.ROOSTER S F-18 - SORTIE 4 116 116 Rooster seethes, outwitted, but concedes the fight... ROOSTER Copy kill.INT.READY ROOM - SORTIE 4 117 117 Everyone exhales, shares a collective look.This is next level shit, even for them.EXT.TARMAC - ELSEWHERE - DUSK 118 118 Close on Rooster, sweating and furious as he does push-ups on the tarmac, punishing himself. (CONTINUED)CHERRY 11.25.19 - OFFICIAL 62. 8FLiX.com FYC SCREENPLAY DATABASE 20221226HONDO Alright. That s enough man.Rooster, that s enough.Hondo pats Rooster on the shoulder. HONDO (CONT D) Tomorrow s another day.Rooster sits up, exhausted.Feet appear next to him.He looks up to see Phoenix above him.PHOENIX What is going on with you? You trying to get kicked out?Breaking the hard deck.Insubordination.That wasn t you up there. Talk to me.What s up?ROOSTER Don t worry about it. PHOENIX I m going on this mission.But if you get kicked out, you could leave us flying with Hangman.So what the hell was that- ROOSTER HE PULLED MY PAPERS. PHOENIX What?Who?ROOSTER Maverick. He pulled my application to the Naval academy. He set me back four years.Phoenix processes.PHOENIX Why would he do that? Rooster does not answer.INT.READY ROOM 119 119 Hangman is staring at something on the wall. |

Table 30: Example 3: Comparison of retrieval results between the SPScanner 1.3B and NV-Embed-v2-7B in the "5 chunks" setting. A portion of the document is displayed, with retrieved sentences highlighted for both models in yellow. Information important for answering the question but missed by NV-Embed-v2-7B is highlighted in red in the text.

| Question: How much younger is Aelis' first husband than Birdy's father? |
| Reference Answer: 32 years. |

| SPScanner 1.3B | NV-Embed |
| --- | --- |
| Attempt: Aelis' first husband is 32 years younger than Birdy's father. | Attempt: Aelis' first husband is approximately 31-51 years younger than Birdy's father. |
| It is I, Birdy.OVER THE IMAGE, A HAND WRITES, AS IF ON AN ILLUMINATED MANUSCRIPT: CATHERINE CALLED BIRDY. INT.STONEBRIDGE MANOR-SOLAR- SAME TIME- MORNING This is Birdy s father, LORD ROLLO S man cave, hung with variously sized antlers and evidence of violent past times. BIRDY (V.O.)I am the Daughter of Lord Rollo.TEXT ON SCREEN: Lord Rollo - 41 years of age- often vain- usually drunk- always greedy (says me) He takes a drink. Then another...(3000 words omitted)...LORD SIDEBOTTOM, Aelis s father, is nearing seventy but still clanking his old bones together in a push chair that rolls between the two seats. TEXT ON SCREEN: LORD GIDEON SIDEBOTTOM - 81 years of age- oldest man in his province- oldest father in England- wears his armour to sleep BERENICE, Aelis s gorgeous young stepmum, looks a thousand times more bored than AISLINN. She is rife with the ennui of entrapment.Aelis leans over the cart s edge and shyly returns Birdy s joyful wave. EXT.STONEBRIDGE MANOR-COURTYARD- MOMENTS LATER- DAY Birdy and Aelis have sequestered themselves gleefully from the grownups on a bench. Aelis bends down behind Birdy, playing with her hair.AELIS Your hair is so long Birdy. You need to brush it.BIRDY I m going to grow it all the way down to my feet...(7000 words omitted)...AELIS Birdy, I am to be married.BIRDY (stricken) To George? AELIS No, to a boy of only nine.George has to marry some horrid old widow named Ethelfritha. And now you will not even be my friend!Aelis rushes out.Birdy looks at the nun wearily. BIRDY (V.O.)For the first time in my life, I am choking on my words. My heart has been shaved and boiled like a parsnip.George is to be married. George is to be married.George.Is.To.Be.Married.Birdy looks at the nun wearily. BIRDY I suppose you re not taking joiners at the convent. | This is Birdy s father, LORD ROLLO S man cave, hung with variously sized antlers and evidence of violent past times. BIRDY (V.O.)I am the Daughter of Lord Rollo.TEXT ON SCREEN: Lord Rollo - 41 years of age- often vain- usually drunk- always greedy (says me) He takes a drink. Then another...(3000 words omitted)...LORD SIDEBOTTOM, Aelis s father, is nearing seventy but still clanking his old bones together in a push chair that rolls between the two seats. TEXT ON SCREEN: LORD GIDEON SIDEBOTTOM - 81 years of age- oldest man in his province- oldest father in England- wears his armour to sleep BERENICE, Aelis s gorgeous young stepmum, looks a thousand times more bored than AISLINN. She is rife with the ennui of entrapment.Aelis leans over the cart s edge and shyly returns Birdy s joyful wave. EXT.STONEBRIDGE MANOR-COURTYARD- MOMENTS LATER- DAY Birdy and Aelis have sequestered themselves gleefully from the grownups on a bench. Aelis bends down behind Birdy, playing with her hair.AELIS Your hair is so long Birdy. You need to brush it.BIRDY I m going to grow it all the way down to my feet...(7000 words omitted)...AELIS Birdy, I am to be married.BIRDY (stricken) To George? AELIS No, to a boy of only nine.George has to marry some horrid old widow named Ethelfritha. And now you will not even be my friend!Aelis rushes out.Birdy looks at the nun wearily. BIRDY (V.O.)For the first time in my life, I am choking on my words. My heart has been shaved and boiled like a parsnip.George is to be married. George is to be married.George.Is.To.Be.Married.Birdy looks at the nun wearily. BIRDY I suppose you re not taking joiners at the convent. |

Table 31: Example 4: Comparison of retrieval results between the SPScanner 1.3B and NV-Embed-v2-7B in the "5 chunks" setting. A portion of the document is displayed, with retrieved sentences highlighted for both models in yellow. Information important for answering the question but missed by NV-Embed-v2-7B is highlighted in red in the text.

### E.8 Performance Change around 128-256k Tokens

As shown in Figure 4 of the main text, SPScanner's performance unexpectedly improves at 128-256k tokens before declining linearly afterward. This counterintuitive pattern occurs because different context length intervals contain distinct document types. In Table 32 below, we show which document types appear at each context length range.

| Token Range | Description and Example Datasets |
|---|---|
| 1–8k tokens | Scientific research papers (qasper, paperqa); legal and nonfiction documents (multifieldqa_en); historical records (2wikimqa, altqa); meeting transcripts focused on speaker identification. |
| 8–16k tokens | Wikipedia passages and screenplays (hotpotqa); government documents (musique); lecture transcripts with multiple-choice questions (coursera); meeting transcripts with relationship queries; historical QA from Wikipedia (altqa_16k). |
| 16–32k tokens | Novels (narrativeqa); legal contracts (legal_contract_qa); passages combined from novels, historical texts, scientific forms, and dictionaries (loogle_CR_mixup_16k, loogle_MIR_mixup_16k, multifieldqa_en_mixup_16k). |
| 32–64k tokens | Combined passages from varied sources (loogle_CR_mixup_32k, loogle_MIR_mixup_32k, multifieldqa_en_mixup_32k); narrative tasks with villain–hero identification (muld_CAC). |
| 64–128k tokens | Combined passages from varied sources (loogle_CR_mixup_64k, loogle_MIR_mixup_64k, multifieldqa_en_mixup_64k); dialogue transcripts requiring character identification across long conversations (longdialogue_qa_eng). |
| 128–256k tokens | Novels with QA or multiple-choice formats (longbook_qa_eng, longbook_choice_eng); financial filings and reports (docfinQA); combined passages from varied sources (loogle_CR_mixup_128k, loogle_MIR_mixup_128k, multifieldqa_en_mixup_128k). |
| 256k+ tokens | Combined passages from novels, historical documents, scientific texts, and dictionaries (loogle_CR_mixup_256k, loogle_MIR_mixup_256k, multifieldqa_en_mixup_256k). |

Table 32: Overview of Datasets by Context Length

### E.9 SPScanner Qualitative Error Analysis

In this section, we provide a qualitative error analysis where GPT-4o Full-Context answers correctly but SPScanner 1.3B's downstream generator answers incorrectly. In most cases, SPScanner fails because it only retrieves fragmented sentences. Table 33 presents three example questions along with their corresponding answers, SPScanner responses, and GPT-4o Full-Context responses.

For the first example, SPScanner correctly retrieved the target sentence "$$MASK$$: (coyly) What a good friend" but missed the crucial preceding context: "CHRISTY: Don't worry.He's in good hands.Trent cracks the door and peers through.The light is dim, but he can make out that they're starting to neck.He closes the door, satisfied." Without this speaker tag establishing Christy's presence in the sleeping compartment, the downstream generator selected the nearest visible female name (Lisa) from the retrieved sentences and answered incorrectly. In contrast, GPT-4o had access to the full contiguous passage showing the complete sleeping-compartment context where only Christy and Trent were present, enabling correct identification.

| Example 1: Dialogue QA ('longdialogue_qa_eng') | |
|---|---|
| **Question** | Which character is $$MASK$$? |
| **Answer** | CHRISTY |
| **SPScanner Answer** | Lisa |
| **GPT-4o Full-Context Answer** | Christy |
| **Example 2: Narrative QA ('longbook_qa_eng')** | |
| **Question** | Why does Sir Dilan loose two boots in London? |
| **Answer** | Angelo needed the scent from an old boot to lure his hound |
| **SPScanner Answer** | Sir Dilan loses two boots in London because they were stolen and used in a plot against him, involving setting a hound on his track. |
| **GPT-4o Full-Context Answer** | Sir Dilan loses two boots in London because Angelo needed an article of his attire to set the hound on his track, leading to the theft of his boots. |
| **Example 3: Document Analysis ('loogle_CR_mixup_16k')** | |
| **Question** | Please select architectures are affected by the Baroque style according to the passage? 1. The Church of Saints Cosme and Damiano. 2. The Church and Convent of São Francisco in Salvador. 3. Portuguese military buildings before 16th century. |
| **Answer** | 1, 2 |
| **SPScanner Answer** | 1 |
| **GPT-4o Full-Context Answer** | 1, 2 |

Table 33: Qualitative Analysis of SPScanner vs. Full-Context GPT-4o.

For the second example, SPScanner's answer was less precise because while its retrieved sentences provided clues about stolen boots being used to set a hound on someone's track, the final answer failed to explicitly identify the antagonist or explain that an old boot was needed specifically for its scent. GPT-4o directly identifies the complete sequence: stealing a new boot, finding it useless for scent-tracking, then stealing an old one that would carry the target's scent. This led to a more complete explanation of why two boots were lost rather than just one.

For the third example, SPScanner failed because it only retrieved fragmented sentences that mentioned baroque features but lacked the complete contextual descriptions needed to properly identify which specific buildings were affected by the Baroque style. While the retrieved sentences contained references like "although the tower is partly baroque" and mentions of baroque ornamentation, the generator couldn't definitively connect these features to both churches.

