# OpenReview forum: "Single-Pass Document Scanning for Question Answering"
_colmweb.org/COLM/2025/Conference — COLM 2025_

### Official Review · Reviewer_jppx · 2025-05-06

**Rating:** 7
**Confidence:** 3
**Ethics Flag:** 1

**Summary:**

The paper introduces a single-pass document scanning approach for question answering, which processes entire documents in linear time while preserving global coherence. The method outperforms chunk-based embedding approaches and competes with large language models at a fraction of the computational cost. Key advantages include better performance than chunk-based methods and lower computational expense compared to full-context models.

**Questions To Authors:**

- The use of binary relevance labels may oversimplify sentence relevance. Could a graded or probabilistic labeling scheme improve performance?
- Why was top-k sentence selection chosen over top-p sampling? Was this decision empirically validated?
- Why not compare with LLMs that support very long contexts (e.g., millions of tokens) for generating synthetic data, besides chunk and pair-based approaches?
- Is the search for connections between chunks performed recursively, or is it limited to nearby context?
- Since the dataset includes sentence-level relevance annotations, why not evaluate retrieval performance directly?
- What explains the performance bounce at 128–256k tokens, coming after a linear drop, and followed by another drop? This pattern seems counterintuitive.

**Reasons To Accept:**

Relevance: The paper addresses a critical challenge in NLP—handling long documents for QA—which is highly relevant to the community.

Methodology: The proposed approach is clear and well-motivated. The methodology is straightforward and well-explained.

Detail and Clarity: The paper provides extensive details, including synthetic data generation, experimental setups, and thorough comparisons with baselines. The presentation is clear and accessible.

**Reasons To Reject:**

Marginal Improvement: The performance gains over vector-based retrievers, while consistent, are relatively modest. A stronger case for the practical impact of these improvements would strengthen the paper.

Missing Retrieval Evaluation: The paper lacks direct evaluation of retrieval performance against other retrievers, which would provide a more comprehensive comparison.

Note: These issues are not severe, and the paper’s strengths outweigh its limitations. Overall, it merits acceptance.

---

> ### Author Response · Authors · 2025-06-02
> **Rebuttal by Authors Part 1**
>
> We appreciate your positive assessment of our work, particularly your recognition that it addresses a critical challenge in NLP and your acknowledgment of the clarity in our methodology and experimental details.
>
> > Marginal Improvement: The performance gains over vector-based retrievers, while consistent, are relatively modest. A stronger case for the practical impact of these improvements would strengthen the paper.
>
> SPScanner is designed for scenarios where a document is introduced for the first time and then queried, demonstrating significant practical advantages over vector-based retrievers in both computational efficiency and accuracy.
>
> Our 130M model achieves higher overall accuracy than all tested embedding baselines while using only a fraction of the computational resources - 19.0 TFLOPs compared to 1287.5 TFLOPs for the most accurate baseline (NV-Embed-v2-7B). Furthermore, our 1.3B variant closes 50% of the performance gap between NV-Embed-v2-7B and GPT-4o Full Context while still using significantly fewer computational resources (197.9 TFLOPs).
>
> These improvements are particularly valuable in resource-constrained environments and applications requiring rapid document processing, where the combination of higher accuracy and reduced computational overhead provides clear practical advantages over existing methods.
>
> >Missing Retrieval Evaluation: The paper lacks direct evaluation of retrieval performance against other retrievers, which would provide a more comprehensive comparison.
>
> > Since the dataset includes sentence-level relevance annotations, why not evaluate retrieval performance directly?
>
> We do not have access to ground-truth annotations of relevant sentences in the 41 long document datasets we used. However, we can use a proxy measure: having GPT-4o annotate relevant sentences while knowing the actual answer to the question, and evaluating both our model and the baselines against this proxy using standard information retrieval metrics such as recall, precision and nDCG. We will report these retrieval evaluation results by the end of the response period.
>
> > The use of binary relevance labels may oversimplify sentence relevance. Could a graded or probabilistic labeling scheme improve performance?
>
> We acknowledge that explicitly graded training labels could potentially capture additional nuances and represent an interesting direction for future work.
>
> > Why was top-k sentence selection chosen over top-p sampling? Was this decision empirically validated?
>
> We chose top-k sentence selection because the approach aligns with established RAG setups ([Xu et al., 2023](https://arxiv.org/abs/2310.03025); [Li et al., 2024](https://arxiv.org/abs/2407.16833)). We will perform an additional experiment using top-p sampling for comparison and provide the results before the end of the discussion period.
>
> > Why not compare with LLMs that support very long contexts (e.g., millions of tokens) for generating synthetic data, besides chunk and pair-based approaches?
>
> While comparing with long-context LLMs would provide valuable insights, this approach would substantially increase per-document processing costs. Given our budget constraints, we chose the current methodology, which enabled us to generate all synthetic data for $1,314 - a feasible cost for us.
>
> > Is the search for connections between chunks performed recursively, or is it limited to nearby context?
>
> The link search is single-pass, not recursive. For each randomly selected 20-sentence chunk, we provide the LLM with a 200-sentence context window (several thousand tokens), in which the initial chunk is located either at the very beginning or the very end of this window. Within that context span, the model identifies meaningful dependencies., and we select the most distant one to formulate a question and label the relevant sentences. Importantly, no newly identified sentences are fed back into the search process.
>
> We apologize for the inaccuracy in Line 134 of the manuscript. It will be corrected with the above clarification.

---

> > ### Author Response · Authors · 2025-06-02
> > **Rebuttal by Authors Part 2**
> >
> > > What explains the performance bounce at 128–256k tokens, coming after a linear drop, and followed by another drop? This pattern seems counterintuitive.
> >
> > Different context length intervals (e.g. 64k-128k, 128k-256k, beyond 256k) contain different document types. Below we summarize the document types at each context length interval:
> >
> > 1-8k tokens: Scientific research papers (qasper, paperqa); legal and nonfiction documents (multifieldqa_en); historical records (2wikimqa, altqa); meeting transcripts focused on speaker identification.
> >
> > 8-16k tokens: Wikipedia passages and screenplays (hotpotqa); government documents (musique); lecture transcripts with multiple-choice questions (coursera); meeting transcripts with relationship queries; historical QA from Wikipedia (altqa_16k).
> >
> > 16-32k tokens: Novels (narrativeqa); legal contracts (legal_contract_qa); passages combined from novels, historical texts, scientific forms, and dictionaries (loogle_CR_mixup_16k, loogle_MIR_mixup_16k, multifieldqa_en_mixup_16k).
> >
> > 32-64k tokens: Combined passages from varied sources (loogle_CR_mixup_32k, loogle_MIR_mixup_32k, multifieldqa_en_mixup 32k); narrative tasks with villain–hero identification (muld_CAC).
> >
> > 64-128k tokens: Combined passages from varied sources (loogle_CR_mixup_64k, loogle_MIR_mixup_64k, multifieldqa_en_mixup 64k); dialogue transcripts requiring character identification across long conversations (longdialogue_qa_eng).
> >
> > 128-256k tokens: Novels with QA or multiple-choice formats (longbook_qa_eng, longbook_choice_eng); financial filings and reports (docfinQA); combined passages from varied sources (loogle_CR_mixup_128k, loogle_MIR_mixup_128k, multifieldqa_en_mixup 128k).
> >
> > 256k+ tokens: Combined passages from novels, historical documents, scientific texts, and dictionaries (loogle_CR_mixup_256k, loogle_MIR_mixup_256k, multifieldqa_en_mixup 256k).

---

> > > ### Comment · Reviewer_jppx · 2025-06-06
> > >
> > > Thank you for the clarifications and the additional information. Please ensure that these points are included in the final version of the paper—particularly the explanation of how your approach uses only a fraction of the computational resources, the new results obtained with top-p, and the observation about different context length intervals containing different document types.

---

> > > > ### Author Response · Authors · 2025-06-07
> > > > **Additional Experiment Results and Interpretation**
> > > >
> > > > Thank you for your reply. We provide additional experiment results below.
> > > >
> > > > > We will report these retrieval evaluation results by the end of the response period.
> > > >
> > > > We now report the retrieval performance of SPScanner. To assess retrieval quality, we provided GPT-4o with the ground-truth answer and asked it to label the sentences that are sufficient to answer each question. These labels were then verified by asking GPT-4o to answer the question using only the retrieved sentences. Of the 5,735 test items, 3,539 received verified sentence annotations. We evaluate on this subset.
> > > >
> > > > SPScanner was compared against other retrieval methods. The results are reported in the following table:
> > > > | Model | Recall@1 | Recall@5 | Recall@10 | Recall@50 | Precision@1 | Precision@5 | Precision@10 | Precision@50 | NDCG@1 | NDCG@5 | NDCG@10 | NDCG@50 |
> > > > |-------|----------|----------|-----------|-----------|-------------|-------------|--------------|--------------|--------|--------|---------|---------|
> > > > | BM25 | 0.1151 | 0.2733 | 0.3516 | 0.5634 | 0.2834 | 0.1582 | 0.1119 | 0.0441 | 0.2834 | 0.2758 | 0.3028 | 0.3674 |
> > > > | Dragon-110M | 0.1246 | 0.2867 | 0.3675 | 0.5954 | 0.3071 | 0.1673 | 0.1169 | 0.0466 | 0.3071 | 0.2930 | 0.3193 | 0.3875 |
> > > > | GTE-Qwen2-1.5B | 0.1417 | 0.3149 | 0.4059 | 0.6445 | 0.3396 | 0.1828 | 0.1290 | 0.0501 | 0.3396 | 0.3219 | 0.3526 | 0.4246 |
> > > > | OpenAI-v3-Large | 0.1509 | 0.3516 | 0.4534 | 0.6822 | 0.3738 | 0.2069 | 0.1440 | 0.0529 | 0.3738 | 0.3579 | 0.3911 | 0.4605 |
> > > > | NV-Embed-v2-7B | 0.1577 | 0.3677 | 0.4629 | 0.6899 | 0.3865 | 0.2145 | 0.1462 | 0.0538 | 0.3865 | 0.3730 | 0.4036 | 0.4733 |
> > > > | SPScanner 130M | 0.1612 | 0.3959 | 0.5133 | 0.7439 | 0.4103 | 0.2409 | 0.1697 | 0.0617 | 0.4103 | 0.4050 | 0.4439 | 0.5157 |
> > > > | SPScanner 1.3B | **0.1943** | **0.4554** | **0.5709** | **0.7899** | **0.4775** | **0.2743** | **0.1900** | **0.0658** | **0.4775** | **0.4668** | **0.5052** | **0.5731** |
> > > >
> > > > SPScanner 1.3B consistently outperforms other retrievers across all metrics, followed by SPScanner 130M. This is consistent with stronger performance in downstream question answering tasks, as evaluated on the 41 benchmark datasets.
> > > >
> > > > > We will perform an additional experiment using top-p sampling
> > > >
> > > > We performed top-p sampling experiments using two parameter settings: p=0.7 and p=1.0. For each setting, we sample sentences until 50 unique sentences are sampled. The results are shown below:
> > > > | Model          | p=0.7 | p=1.0 | Retrieving top 50 sentences |
> > > > |----------------|-------|-------|----------------------------|
> > > > | SPScanner 130M | 59.1  | 58.0  | 60.0                       |
> > > > | SPScanner 1.3B | 60.3  | 59.2  | 61.8                       |
> > > >
> > > > Both top-p sampling configurations (p=0.7 and p=1.0) underperform the top-50 sentence retrieval setting.

---

> > > > > ### Comment · Reviewer_jppx · 2025-06-09
> > > > >
> > > > > Thanks for the additional results! I believe the retrieval-related findings are very relevant and make the paper even stronger. Please make sure to include them in the final version of the paper.

---

### Official Review · Reviewer_t7hJ · 2025-05-12

**Rating:** 7
**Confidence:** 3
**Ethics Flag:** 1

**Summary:**

This paper presents a study on adapting the state-space model to the task of long document question answering (QA).  This is a generally well written paper. The authors propose an interesting synthetic data generation strategy from large LLMs that is based on document connections between entities and themes. The proposed approach consistently outperforms more expensive models on most QA tasks considered while being more efficient. The approach described in this paper is a good step towards developing more efficient long document QA systems, however I have concerns regarding experimental results such as comparison to NV-embed on conversational, and on writing clarity (see Reasons to Reject).

**Questions To Authors:**

1. In Table 4 you combine different data generation strategies. Have you experimented with combining those strategies?

**Reasons To Accept:**

* Interesting link-based data generation approach
* Outperforms more expensive baselines on most datasets
* Thorough analysis and ablations
* Open-source code

**Reasons To Reject:**

* Lacks discussion on why NV-embed outperforms this method on conversational datasets. Is the data generation strategy followed here not suitable to conversational tasks?
* Lacks discussion on what is missing to get to the performance of the big LLM (GPT-4o).
* Lacks discussion on why link-based synthetic data is more suitable for the SSM than the baselines (end of Section 6).
* Writing clarity could be improved. For instance, references to the Appendix in Sections 4.1 and 7.3 should also provide the main conclusion so that the main text is complete.

---

> ### Author Response · Authors · 2025-06-02
> **Rebuttal by Authors Part 1**
>
> Thank you for your detailed review and for your assessment that link-based synthetic data generation is interesting, that SPScanner demonstrates performance advantages over more expensive baselines, and that our analysis and ablations are thorough.
>
> > Lacks discussion on why NV-embed outperforms this method on conversational datasets. Is the data generation strategy followed here not suitable to conversational tasks?
>
> We thank the reviewer for this important observation. NV-embed’s superior performance on conversational datasets can be attributed to the specific nature of these tasks, which often do not require capturing long-range dependencies from distant parts of the document. Since embedding models can retrieve local context but falter at long-range dependencies, these conversational datasets avoid the weakness of embedding models.
>
> Of the 707 data points in the conversational datasets, 400 focus specifically on speaker identification tasks, divided into two categories: 200 questions about masked name entities and 200 requiring identification of speakers for given sentences. Both tasks only require knowing the immediate context to answer the question. We provide one example for each task below.
>
> Masked Name Entities Task:
>
> Question: “Which character is \\$\\$MASK\\$\\$?”
>
> Context:
> PATRICK: I do not care for your language.
> LEONARD: I'm only trying to help you guys.
> Leonard stops exercising.
> \\$\\$MASK\\$\\$: (CONT'D) You're fat dad. Mom is fat. Us kids are fat. (CONTINUED).
> CONTINUED:
> PATRICK: What's your point?
>
> Answer: LEONARD
>
> Speaker Identification Task:
>
> Question: “‘Who would like to come in on that? Go on, Catherine.’ said by ___”
>
> Context:
> "And Catherine, I just wonder, extending Alun's point, ..." said by Huw Irranca-Davies AM
> "On the point about being boring, generally boring is good as far as the EU is concerned ..." said by Alun Davies AM
> "Who would like to come in on that? Go on, Catherine." said by ___
>
> Answer: Huw Irranca-Davies AM
>
> We will include this analysis in the revised manuscript.

---

> > ### Author Response · Authors · 2025-06-02
> > **Rebuttal by Authors Part 2**
> >
> > > Lacks discussion on what is missing to get to the performance of the big LLM (GPT-4o).
> >
> > We will include a qualitative error analysis where GPT-4o Full context answers correctly but SPScanner’s downstream generator answers incorrectly in the revised manuscript. In most cases, SPScanner fails because it only retrieves fragmented sentences. Here are three such examples, each with an error analysis:
> >
> > Example 1 (from longdialogue_qa_eng):
> >
> > Question: Which character is \\$\\$MASK\\$\\$?
> >
> > Answer: CHRISTY
> >
> > SPScanner Answer: Lisa
> >
> > GPT-4o Full Context Answer: Christy
> >
> > SPScanner correctly retrieved the target sentence “\\$\\$MASK\\$\\$: (coyly) What a good friend” but missed the crucial preceding context: “CHRISTY: Don't worry.He's in good hands.Trent cracks the door and peers through.The light is dim, but he can make out that they're starting to neck.He closes the door, satisfied.” Without this speaker tag establishing Christy’s presence in the sleeping compartment, the downstream generator selected the nearest visible female name (Lisa) from the retrieved sentences and answered incorrectly. In contrast, GPT-4o had access to the full contiguous passage showing the complete sleeping-compartment context where only Christy and Trent were present, enabling correct identification.
> >
> > Example 2 (from longbook_qa_eng)
> >
> > Question: Why does Sir Dilan loose two boots in London?
> >
> > Answer: Angelo needed the scent from an old boot to lure his hound
> >
> > SPScanner Answer: Sir Dilan loses two boots in London because they were stolen and used in a plot against him, involving setting a hound on his track.
> >
> > GPT-4o Full Context Answer: Sir Dilan loses two boots in London because Angelo needed an article of his attire to set the hound on his track, leading to the theft of his boots.
> >
> > SPScanner’s answer was less precise because while its retrieved sentences provided clues about stolen boots being used to set a hound on someone’s track, the final answer failed to explicitly identify the antagonist or explain that an old boot was needed specifically for its scent. GPT-4o directly identifies the complete sequence: stealing a new boot, finding it useless for scent-tracking, then stealing an old one that would carry the target’s scent. This led to a more complete explanation of why two boots were lost rather than just one.
> >
> > Example 3 (from loogle_CR_mixup_16k)
> >
> > Question:
> > Please select architectures are affected by the Baroque style according to the passage?
> > 1. The Church of Saints Cosme and Damiano.
> > 2. The Church and Convent of S?o Francisco in Salvador.
> > 3. Portuguese military buildings before 16th century.
> >
> > Answer: 1, 2
> >
> > SPScanner Answer: 1
> >
> > GPT-4o Full Context Answer: 1, 2
> >
> > SPScanner failed because it only retrieved fragmented sentences that mentioned baroque features but lacked the complete contextual descriptions needed to properly identify which specific buildings were affected by the Baroque style. While the retrieved sentences contained references like “although the tower is partly baroque” and mentions of baroque ornamentation, the generator couldn’t definitively connect these features to both churches.
> >
> > > Lacks discussion on why link-based synthetic data is more suitable for the SSM than the baselines (end of Section 6).
> >
> > Link-based synthetic data is designed to help models capture long-range dependencies between distant parts of the document. Baseline embedding models have short context windows that cannot capture such long-range dependencies. In contrast, SSMs can process the entire long document in a single pass to better capture the long-range dependencies in link-based data. We will add this discussion to the revised manuscript.
> >
> > We provide examples of link-based synthetic data in Table 18 of Appendix B.4.
> >
> > > Writing clarity could be improved. For instance, references to the Appendix in Sections 4.1 and 7.3 should also provide the main conclusion so that the main text is complete.
> >
> > We will revise the main text of the manuscript to improve clarity. Specifically, we will update Sections 4.1 and 7.3 to include main conclusions from referenced appendices.
> >
> > > In Table 4 you combine different data generation strategies. Have you experimented with combining those strategies?
> >
> > We experimented with mixing link-based data with chunk-based data. Specifically, we train the 1.3B model with 400k link-based examples combined with 600k chunk-based examples (totaling one million training examples). This mixed training approach resulted in degraded performance on our validation set, which was drawn from 8 benchmark tasks and kept disjoint from test sets (Lines 169-171). Because of such degraded performance, no further combinations are experimented.
> >
> > We did not experiment with including pair-based data in the mixture, given that the link-based generation strategy already improves upon the pair-based strategy.

---

> > > ### Comment · Reviewer_t7hJ · 2025-06-08
> > >
> > > Thank you for your response, please revise the manuscript accordingly. I have increased my score.

---

### Official Review · Reviewer_gFrf · 2025-05-13

**Rating:** 7
**Confidence:** 3
**Ethics Flag:** 1

**Summary:**

The paper introduces Single‑Pass Document Scanning (SPScanner), a state‑space‑model‑based retriever that reads an entire long document once, scores every sentence for relevance to a question, and forwards only the top‑k sentences to a generator LLM. Built on the Mamba‑2 architecture, the scanner runs in linear time and is trained with a new link‑based synthetic‑question pipeline that teaches it to capture long‑range dependencies. Across 41 long‑document QA benchmarks—spanning educational, creative, official and conversational texts—SPScanner (130 M & 1.3 B) outperforms strong transformer‑based embedding retrievers (e.g., NV‑Embed‑v2‑7B, Stella‑1.5B) and even approaches GPT‑4o’s full‑context accuracy on 256 k‑token inputs, while using fewer FLOPs and comparable latency.

**Questions To Authors:**

* What specific characteristics of the link‑based questions (e.g., span distance, discursive cues, entity co‑reference) do you believe most strongly encourage the model to learn long‑range dependencies?

**Reasons To Accept:**

* Clear motivation &  Technical contribution – chunk‑based RAG loses global context, full‑context LLMs are quadratic; framing retrieval as linear‑time sentence scoring is well motivated. The authors adapts Mamba‑2 SSM with a sentence‑level classification head, enabling autoregressive global context use without attention.
* Novel link‑based synthetic data – generates questions that require distant evidence, producing stronger long‑range reasoning than chunk‑ or pair‑based baselines (+2‑8 pp accuracy).
* Comprehensive evaluation – 41 benchmarks, 5  737 QA instances up to 390 k tokens; consistent average gains of 2‑5 pp over best embedding retrievers and parity with GPT‑4o on very long docs.

**Reasons To Reject:**

* Limited architectural novelty – core model is a straightforward repurposing of Mamba‑2 with a binary head; advances over prior SSM retrievers (Hydra, Monarch Mixer) are incremental.

---

> ### Author Response · Authors · 2025-06-02
> **Rebuttal by Authors**
>
> We appreciate your comprehensive evaluation of our work and your assessment that it has clear motivation, technical contribution, and novelty of link-based synthetic data.
>
> > Limited architectural novelty – core model is a straightforward repurposing of Mamba‑2 with a binary head; advances over prior SSM retrievers (Hydra, Monarch Mixer) are incremental.
>
> Our primary novelty lies in determining how to effectively train an SSM model as a single-pass, discriminative scanner for identifying relevant information in long documents.
>
> Straightforward adaptations of existing approaches fail to achieve similar performance. Training Mamba for generative retrieval achieves only 33.5% accuracy (Appendix E.2), while direct question answering achieves only 27.6% (Appendix E.3). Additionally, straightforward synthetic data approaches like chunk-based (using individual document segments) and pair-based (matching similar chunks) produce questions that don't require understanding document-wide connections and lead to worse performance (Table 4).
>
> The effectiveness of our approach is demonstrated by: (1) the model outperforms embedding models 8-60x larger while using fewer computational resources (Table 1 and 2), (2) fine-tuning these embedding models on our synthetic data shows no improvement (Line 270-273), and (3) we observe strong performance on documents up to length 256k, which is more than 20x longer than any documents observed during training (Line 264-269).
>
> > What specific characteristics of the link‑based questions (e.g., span distance, discursive cues, entity co‑reference) do you believe most strongly encourage the model to learn long‑range dependencies?
>
> We believe that the semantic relationship between linked chunks is a primary driver for learning long-range dependencies. If the span distance between linked chunks is long (e.g. 200 sentences apart rather than 20 sentences apart), it is possible such a natural long-range relationship (e.g., recurring mention of characters in a novel) would better encourage the model to learn long-range dependencies.

---

> > ### Author Response · Authors · 2025-06-07
> > **Additional Experiment Results and Interpretation**
> >
> > > Limited architectural novelty – core model is a straightforward repurposing of Mamba‑2 with a binary head; advances over prior SSM retrievers (Hydra, Monarch Mixer) are incremental.
> >
> > We evaluate M2-BERT, a retrieval-optimized version of Monarch Mixer, on our test set (5735 data points) using 5-chunk retrieval. We use the version of M2-BERT which is served by Together AI. The results are reported in the following table:
> > | Model              | Educational (n=1967) | Creative (n=1733) | Official (n=1328) | Conversational (n=707) | Average Accuracy |
> > |--------------------|----------------------|-------------------|-------------------|------------------------|------------------|
> > | M2-BERT (5 chunks) | 52.3                 | 27.6              | 37.8              | 34.9                   | 38.2             |
> > | SPScanner 130M     | 70.4                 | 54.1              | 59.5              | 49.5                   | 60.0             |
> >
> > SPScanner 130M significantly outperforms M2-BERT. As an embedding model, M2-BERT encodes each 300-word window in isolation, and cannot utilize cross-chunk cues. SPScanner evaluates every sentence in full-document context and is trained on link-based synthetic questions that reward long-range evidence retrieval. These architectural and data differences account for the 21-point average accuracy margin.

---

> > > ### Comment · Reviewer_gFrf · 2025-06-08
> > >
> > > Thank you for the response. That answered my question

---

### Official Review · Reviewer_Dc8t · 2025-05-13

**Rating:** 6
**Confidence:** 3
**Ethics Flag:** 1

**Summary:**

The paper introduces Single-Pass Document Scanning (SPScanner), a novel approach for long-document question answering (QA) that addresses the limitations of traditional methods like chunk-based embedding models (which lose global context) and full-context transformers (which are computationally expensive). SPScanner leverages a state-space model architecture (Mamba-2) to process entire documents linearly and identify query-relevant sentences. And these sentences are then used to generate final answer. SPScanner can achieve better performance than embedding based retrievers and be much faster than full-context methods.

**Questions To Authors:**

1. In section 5.5, you finetune the embedding models with 1 million link-based synthetic data. But during inference, the documents are processed in chunks of fixed-length 300 words, which may cause unfinished sentences in the chunk. Will this gap hinders the performace of finetuned embedding models? (The FT models only show little improvement in Table 1)
2. Is there any ablation study about the number of selected sentences? Since there is still some gap between SPScaner and GPT-4o Full Context, will more selected sentences reduce this gap?

**Reasons To Accept:**

1. SPScanner achieves better performance and speed compared to RAG methods.
2. The link-based synthetic data generation method can help construct high quality data.
3. The paper's experimental design and results are thorough and well-executed.

**Reasons To Reject:**

1. The novelty of this paper is limited, since such sentence extraction method has been widely utilized in open domain question answering[1][2]. Besides there is no discussion about these methods in related work.

[1] RECOMP: Improving retrieval-augmented LMs with context compression and selective augmentation. *ICLR 2024*

[2] FastFiD: Improve Inference Efficiency of Open Domain Question Answering via Sentence Selection. *ACL 2024*

---

> ### Author Response · Authors · 2025-06-02
> **Rebuttal by Authors**
>
> Thank you for your thorough review and for recognizing both the performance advantages of SPScanner over RAG methods and the quality of our link-based synthetic data generation approach.
>
> > The novelty of this paper is limited, since such sentence extraction method has been widely  utilized in open domain question answering[1][2]. Besides there is no discussion about these methods in related work...
>
> We appreciate the pointer to RECOMP and FastFid and will add a paragraph in the Related Work section explicitly comparing our approach to these sentence-extraction methods. We would like to evaluate these models on our tasks. However, these methods rely on in-domain fine-tuning for each specific dataset, whereas SPScanner uses the same checkpoint for all 41 datasets. We also do not use any of the train sets of these 41 datasets for training SPScanner.  Since only RECOMP provided a checkpoint (which has been finetuned for HotpotQA using its train set), this checkpoint cannot be fairly evaluated on these 41 datasets.
>
> Our contributions are distinct in two key ways. First, RECOMP and FastFiD employ transformer-based approaches that scale quadratically with input sequence length. In contrast, our SPScanner leverages a linear-time state space model, enabling efficient processing of extremely long documents that would be much more expensive for these existing methods.
>
> Secondly, we introduce a novel link-based synthetic supervision signal that trains the model to identify and utilize sparse, interconnected information across the given long document. The idea to construct synthetic data in this way is fundamentally different from using the in-domain train sets to fine-tune models for specific test sets.
>
> > In section 5.5, you finetune the embedding models with 1 million link-based synthetic data. But during inference, the documents are processed in chunks of fixed-length 300 words, which may cause unfinished sentences in the chunk. Will this gap hinders the performace of finetuned embedding models? (The FT models only show little improvement in Table 1)
>
> In Appendix A.4, we compare different setups (e.g. retrieve 50 sentences or retrieve 5 chunks of 300 words) for embedding models. Embedding models’ setup to process documents in chunks of 300 words is chosen because this setup often elicits the best performance from embedding models. For the fine-tuned embedding model, our experiment shows the 5-chunk setup is comparable to the 50 sentences setup:
> | Model              | Retrieving 50 sentences | Retrieving 5 chunks |
> |--------------------|-------------------------|---------------------|
> | Contriever-110M-FT| 54.8                    | 54.8                |
>
> Before the end of the discussion period, we will perform this  additional experiment: We will transform the linked-based data into 300-word chunks and fine-tune the embedding model on the transformed training data. We will then evaluate the fine-tuned model with the 5 300-word chunks setup.
>
> > Is there any ablation study about the number of selected sentences? Since there is still some gap between SPScaner and GPT-4o Full Context, will more selected sentences reduce this gap?
>
> The 10-sentence retrieval results are provided in Appendix A.4 and A.5. Results show that retrieving 50 sentences consistently outperforms retrieving 10 sentences:
> | Model          | Retrieving 10 sentences | Retrieving 50 sentences |
> |----------------|-------------------------|-------------------------|
> | SPScanner 130M | 51.8                    | 60.0                    |
> | SPScanner 1.3B | 54.1                    | 61.8                    |

---

> > ### Author Response · Authors · 2025-06-07
> > **Additional Experiment Results and Interpretation**
> >
> > > Before the end of the discussion period, we will perform this additional experiment: We will transform the linked-based data into 300-word chunks and fine-tune the embedding model on the transformed training data. We will then evaluate the fine-tuned model with the 5 300-word chunks setup.
> >
> > We evaluated the effect of sentence vs. chunk-based grouping on embedding model performance. Our experiments separately varied the grouping method during finetuning and at test-time. We fine-tuned two embedding models, Contriever-110M and GTE-Qwen2-1.5B. The results are given in the following table:
> > | Model                        | Retrieving 50 sentences | Retrieving 5 chunks |
> > |------------------------------|-------------------------|---------------------|
> > | Contriever (sentence FT)     | 54.8                    | 54.8                |
> > | Contriever (chunk FT)        | 49.2                    | 52.2                |
> > | GTE-Qwen2-1.5B (sentence FT) | 54.1                    | 55.8                |
> > | GTE-Qwen2-1.5B (chunk FT)    | 53.3                    | 53.0                |
> >
> > The results show that finetuning the models to retrieve sentences generally outperforms finetuning to retrieve chunks, independent of the test-time retrieval method. There may also be a trend towards chunk-based retrieval (at test time) outperforming sentence retrieval, though this appears to be a weaker effect.

---

> > > ### Comment · Reviewer_Dc8t · 2025-06-10
> > >
> > > Thank you for your response. I think your response addressed my questions. Since my original score is already positive, I will keep my score.

---

> ### Author Response · Authors · 2025-06-09
> **Response requested**
>
> Dear Reviewer Dc8t,
>
> Thank you again for your thorough review of our paper. Since the discussion period is coming to an end soon, we would greatly appreciate it if you could indicate whether our rebuttal addresses the questions raised in your initial review.
>
> Thank you for your time,
>
> SPScanner Authors

---

### Author Response · Authors · 2025-06-11
**Summary of Discussion Period**

We thank the reviewers for their detailed feedback. We are pleased that all reviewers recognized the importance of our work in addressing long-document QA. They agreed that our work, which "frames retrieval as linear-time sentence scoring" is "well motivated" (reviewers gFrf, and jppx). Reviewer Dc8t praised our approach as "a novel approach for long-document question answering (QA) that addresses the limitations of traditional methods," and reviewer t7hJ commented that our work is a "good step towards developing more efficient long document QA systems." Furthermore, reviewer jppx noted that it addresses a "critical challenge in NLP.”.

Our responses focused on two main areas:

First, we clarified that the paper's core novelty lies in determining how to effectively train an SSM model as a single-pass scanner for identifying relevant information in long documents. We demonstrated that our link-based synthetic data generation is fundamentally different from existing methods. Both reviewers gFrf and t7hJ recognized our synthetic data approach as "novel" and "interesting".

Second, we conducted new experiments during the rebuttal period. To address feedback from reviewer jppx, we performed a direct evaluation of retrieval performance, showing our SPScanner models consistently outperform other retrieval method on long documents. Reviewer jppx found these new results "very relevant" and stated they "make the paper even stronger". Furthermore, inspired by comments from reviewer gFrf, we added a comparison against M2-BERT, showing our model achieves a 21-point gain in average accuracy, and reviewer gFrf noted that these experiments resolved their concerns. We also provided additional ablation studies on sentence selection and finetuning strategies, and reviewer Dc8t stated the "response addressed my questions".

The revisions have created reviewer consensus around our key contribution: providing an efficient single-pass document scanning method for long-document question answering. The model's ability to preserve global coherence allows it to consistently outperform strong chunk-based embedding methods and compete with large language models at a fraction of the computational cost.

---

### Decision · Program_Chairs · 2025-07-08

**Decision:**

Accept

**Comment:**

The authors and the reviewers engaged in useful discussions, which has led to a stronger paper already. I vote to accept the paper, and expect the authors to update the paper with the new results and revisions.